# Faithful Image Editing via Degraded Representations

## Abstract

Rectified flow and diffusion-based models currently represent the state-of-the-art in image editing, leveraging powerful pre-trained generative priors to produce visually compelling modifications. Despite their impressive capabilities, maintaining faithfulness to the source image – preserving structure and photometric characteristics while satisfying a target prompt – remains a persistent challenge in this domain. Direct traversal between source and target distributions in rectified flow frameworks offers a promising direction for improving fidelity. However, identifying trajectories that are both semantically effective and strictly structure-preserving remains an open problem. In this work, we propose an optimization- and inversion-free image editing framework for rectified flow models, such as SD3 and FLUX.1-dev. Our central insight is to operate within a carefully designed degraded representation space that constrains editing trajectories and suppresses unintended collateral modifications to the target. We first illustrate the potential of such degraded representations for generative-prior-based editing and then develop a method to project editing trajectories onto this space. The resulting method, Editing via Degraded Representations (EDR), systematically eliminates unfaithful trajectory deviations while preserving the flexibility required to satisfy the target text prompt. Extensive quantitative and qualitative evaluations demonstrate that EDR achieves precise, high quality edits that offer superior fidelity with negligible restriction in editability, establishing a new state-of-the-art in faithful image editing. *Code will be released upon acceptance.*

## 1 Introduction

Diffusion models Sohl-Dickstein et al. (2015); Ho et al. (2020) and Rectified Flow (RF) models Liu et al. (2023); Esser et al. (2024) have revolutionized image synthesis. Leveraging these powerful generative priors has consequently emerged as a dominant paradigm for image editing Kulikov et al. (2025); Brack et al. (2024); Mokady et al. (2023); Rout et al. (2025), enabling flexible and photorealistic manipulations grounded in the strong structural representations learned by diffusion and RF models Rombach et al. (2022); Esser et al. (2024).

Despite their impressive generative capabilities, existing approaches often struggle to produce faithful edits, particularly when operating on real images Garibi et al. (2024). Faithful editing requires preserving the global structure and color distribution of the source image while restricting modifications strictly to those necessary for satisfying the target prompt. A primary line of research Kawar et al. (2023); Valevski et al. (2023); Zhang et al. (2023) addresses this challenge through test-time optimization, adapting the generative model to the source image prior for editing. Although this strategy achieves high fidelity, it incurs substantial computational overhead due to per-image optimization, limiting its practicality for large-scale or real-time applications.

Addressing that, a second line of work explores optimization-free approaches Meng et al. (2021); Mokady et al. (2023); Tumanyan et al. (2023). Early methods along this line injected random noise into the source image followed by denoising for editing. This frequently resulted in significant drift and an unstable fidelity–editability trade-off Meng et al. (2021). To reduce this drift, inversion techniques were introduced to recover the latent noise that reconstructs the source image via a deterministic ordinary differential equation (ODE) trajectory. While standard DDIM inversion Song et al. (2020) avoids explicit optimization, numerical

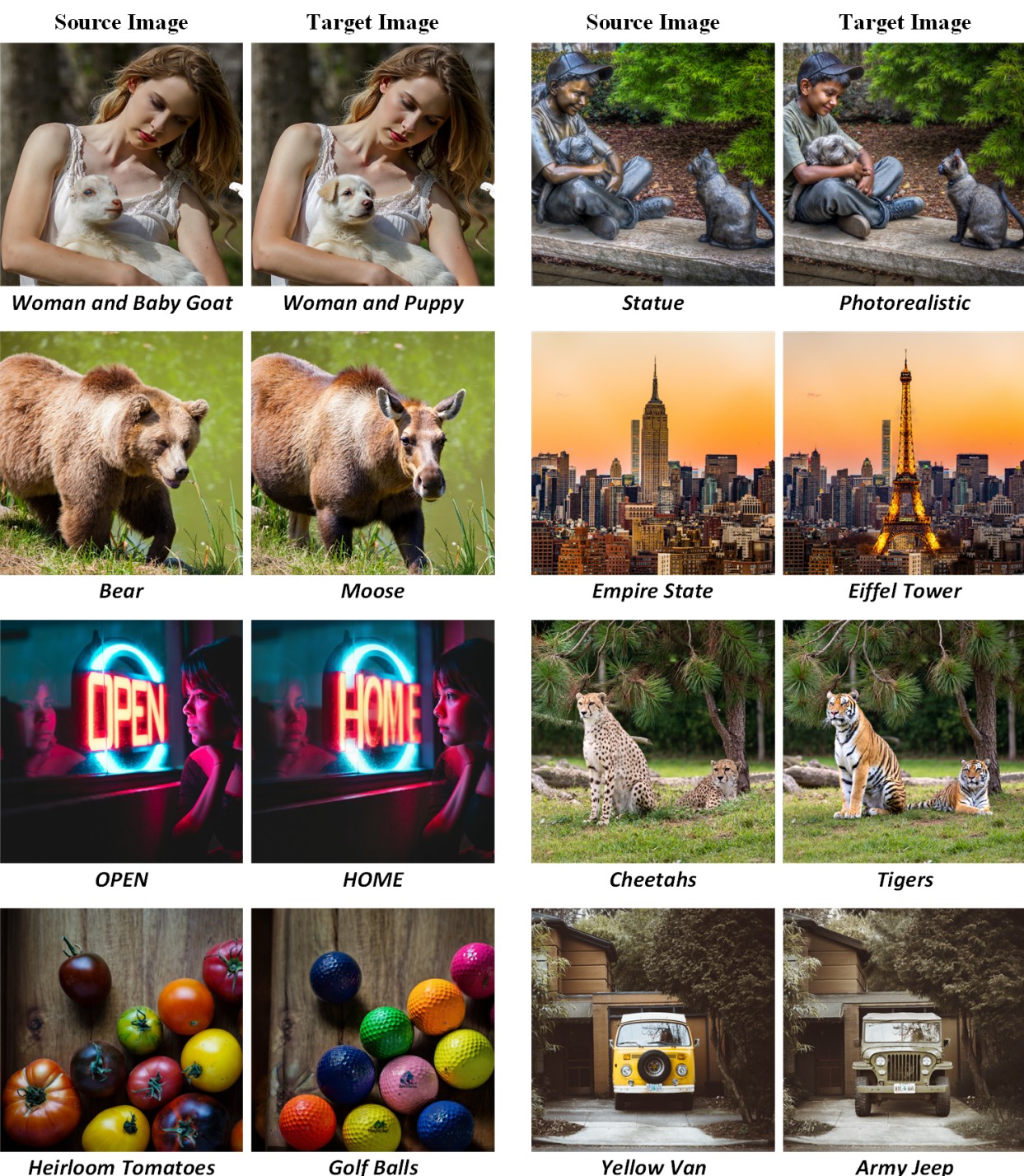

Figure 1: Editing samples using the proposed method on FLUX.1-dev (top two rows) and SD3 (bottom two rows). Our approach projects the editing trajectories onto degraded representations that are invariant to insignificant intensity and structural variations. The resulting edits effectively prevent unintended side effects. Note that no explicit or implicit masks are used in our method.

discretization errors accumulate, often leading to structural inconsistencies during editing. Subsequent works proposed mathematically exact inversion schemes Wallace et al. (2023); Pan et al. (2023) or straight-line RF trajectories Lu et al. (2023); Esser et al. (2024) to improve consistency. However, the exact inverted noise corresponding to the source image may conflict with the target prompt, resulting in weak or unnatural edits Huberman-Spiegelglas et al. (2024). More recently, FlowEdit Kulikov et al. (2025) introduced an inversion-free framework that infers a direct trajectory between the source and target distributions. Although this formulation reduces transition cost and improves fidelity relative to inversion-based methods, it does not guarantee that the inferred trajectory strictly preserves the desirable content. Moreover, to stabilize the editing path, it requires aggregating multiple model predictions at each time step, increasing inference complexity.

In this work, we investigate the editing via generative prior from a trajectory-centric perspective to enable effective yet faithful image editing (see Figure 1). Editing generative trajectories in latent space can traverse multiple directions while still satisfying a target prompt. While some of these directions correspond to semantically meaningful transformations, others introduce unintended variations arising from stochasticity or entangled latent representations shaped by spurious correlations. We show that such trajectories can be systematically constrained by projecting them onto a degraded representation where selected characteristics are suppressed or rendered invariant. We illustrate our key intuition in Figure 2. Consider the representations of an image (red square) and one of its edited versions (blue circle). When mapped to degraded forms—such as edge-based or blurred representations—specific attributes are removed: edge representations discard intensity information, whereas blurred representations suppress fine structural details. In this degraded space, trajectories contain none or minimal components along the directions corresponding to the suppressed characteristics. Consequently, projecting editing trajectories onto degraded trajectories enables effective removal of latent components associated with these characteristics, thereby eliminating unintended deviations.

Building upon this insight, we propose Editing via Degraded Representations (EDR), a method that approximates the projection of editing trajectories onto a tailored degraded space. Our degraded representation combines Gaussian structural smoothing with dynamic range reduction, inducing invariance to subtle structural perturbations as well as incidental intensity and color shifts that frequently arise during editing. By constraining trajectories within this space, EDR enforces high-fidelity edits that preserve the foundational structure and color statistics of the source image. At the same time, the degraded representation remains sensitive to substantial semantic modifications when required by the target prompt, ensuring editability is not unduly restricted.

Our contributions are summarized as follows:

- We introduce a mechanism for constraining generative editing trajectories via projection onto *degraded representations*, thereby suppressing entangled and spurious latent directions.

- We propose Editing via Degraded Representations (EDR) method, which maps editing trajectories to a custom degraded space combining *Gaussian spatial smoothing* and *dynamic range reduction*, achieving invariance to unintended structural and intensity shifts, respectively.

- We conduct extensive quantitative and qualitative evaluations demonstrating that EDR achieves highly faithful edits and outperforms state-of-the-art methods maintaining remarkable source fidelity with negligible restriction in editability.

## 2 Related Work

Our work relates to diffusion- and flow-based image editing, and is particularly relevant to controlled editing with rectified flow. Below, we discuss this relevance, contextualizing our contribution with respect to recent advances.

**Rectified Flow Models.** ODE-based generative formulations are increasingly replacing stochastic diffusion processes as the dominant paradigm in generative modeling. Rectified Flow (RF) Liu et al. (2023)

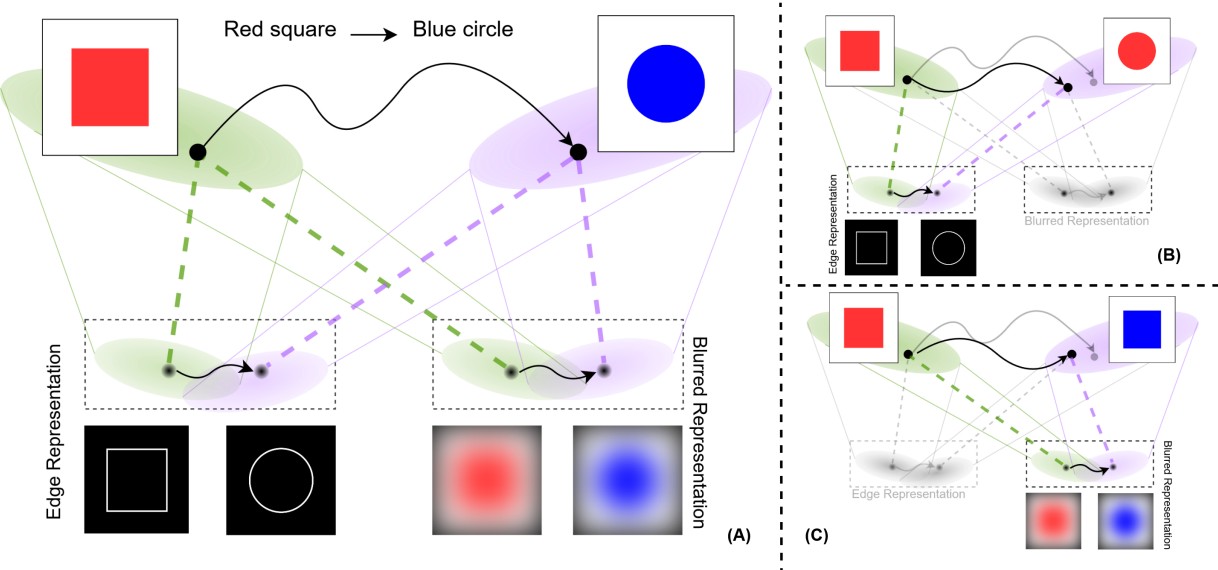

Figure 2: Illustration of our key insight for faithful image editing via degraded representation. **(A)** A source image (red square) and its target edited version (blue circle) get mapped onto their degraded (edge and blurred) representations. The edge representation is inherently invariant to color changes, while the blurred representation is largely insensitive to structural changes. **(B)** The editing trajectory gets projected onto the edge-based degraded representation, which ignores color variations. This prevents the editing process from altering the colors of the source image. **(C)** The editing trajectory is projected onto the blurred degraded representation. Since this representation is largely insensitive to structural information, the resulting trajectory does not modify the structure, thereby the resulting output retains the structure of the source image. Note that this figure conceptually illustrates the principle of editing via degraded representations and does not depict our method exactly; in particular, the edge representation is shown only for illustration and is not used in the deployed method (Section 3.2).

introduced a deterministic transport perspective, learning straightened trajectories between source and data distributions. Related frameworks, including Flow Matching Lipman et al. (2022) and Stochastic Interpolants Albergo et al. (2025), similarly focus on directly learning vector fields that define probability transport paths. Compared to conventional diffusion models Sohl-Dickstein et al. (2015); Song & Ermon (2019); Ho et al. (2020), these approaches offer improved sampling efficiency and more stable training dynamics. However, while rectified flows provide cleaner generative trajectories, they do not inherently address the problem of trajectory control during editing; namely, how to restrict the path to preserve specific structural and photometric properties of a given image. In the context of rectified flow models, our work directly addresses this gap by introducing a mechanism for constraining editing the trajectories.

**Diffusion- and Flow-Based Image Editing.** Diffusion-based editing leverages powerful pre-trained generative priors to enable realistic and semantically aligned modifications Rombach et al. (2022). Existing methods largely fall into three categories: training-based methods, test-time optimization, and optimization-free strategies.

*Training-based methods* Brooks et al. (2023); Wasserman et al. (2025) learn dedicated editing networks that produce fast and faithful modifications. While effective, they rely on synthetic supervision and incur substantial training costs. Even reduced-resource alternatives Zhang et al. (2025) remain dependent on additional learning procedures. In contrast, our method requires neither retraining nor synthetic supervision, operating directly with pre-trained models. *Test-time optimization approaches* Kawar et al. (2023); Valevski et al. (2023) adapt the generative model to each source image to enhance faithfulness. Although this yields high-quality edits, it introduces significant inference latency due to per-image fine-tuning. Our method avoids

per-instance optimization entirely while maintaining high structural and photometric fidelity. *Optimization-free approaches* attempt to balance efficiency and faithfulness but often lack principled trajectory control. Attention-injection methods Tumanyan et al. (2023); Hertz et al. (2022); Alaluf et al. (2024); Patashnik et al. (2023); Hertz et al. (2024) guide generation through semantic feature manipulation, yet they do not explicitly regulate latent transport directions, leaving edits susceptible to unintended structural or color drift. Mask-based strategies Couairon et al. (2022); Avrahami et al. (2023) spatially restrict edits but do not constrain global distributional shifts. Inversion-based techniques Song et al. (2020); Huberman-Spiegelglas et al. (2024); Garibi et al. (2024); Mokady et al. (2023); Wallace et al. (2023) recover the latent noise corresponding to the source image to reduce drift. However, exact inversion often conflicts with target prompts, resulting in weak edits, while approximate inversion accumulates discretization errors that degrade structure.

Recent flow-based editors such as RF Edit Wang et al. (2024) and FlowEdit Kulikov et al. (2025) advance toward Flow Matching formulations. FlowEdit Kulikov et al. (2025) removes explicit inversion by directly coupling source and target vector fields using the ODE structure of flow models Liu et al. (2023). This significantly improves structural fidelity relative to diffusion-based inversion methods. Nevertheless, stabilizing the learned trajectory requires aggregating multiple model predictions at each time step, increasing computational cost. More fundamentally, these methods aim to find a suitable trajectory but do not explicitly suppress undesirable latent directions during transport.

In contrast, our approach introduces an orthogonal perspective: rather than refining inversion, modifying attention, or averaging vector fields, we constrain editing trajectories through projection onto a carefully designed degraded representation. This mechanism explicitly removes components associated with unintended structural and photometric variations, enabling faithful edits without retraining, per-image optimization, or multi-step aggregation.

**Rectified Flows in Large-Scale Generative Modeling.** Recent large-scale text-to-image systems, including Stable Diffusion 3 (SD3) Esser et al. (2024) and FLUX Labs et al. (2025), adopt rectified flow formulations, demonstrating superior sample quality, improved prompt alignment, and scalable training dynamics. Beyond image synthesis, RF-based methods have been extended to unsupervised domain translation Wang et al. (2024), video generation Davtyan et al. (2023); Ifriqi et al. (2025); Jin et al. (2024), and inverse problem solving Zhu et al. (2024). While these works establish rectified flows as a powerful generative backbone, they do not address controlled editing under strict fidelity constraints. Our work builds upon the strengths of rectified flows while introducing an explicit trajectory-constraint mechanism tailored to faithful image editing.

## 3 Methodology

Image editing with diffusion and RF models enables flexible transformation between source and target concepts by navigating learned generative trajectories. However, these editing trajectories frequently deviate along unintended latent directions, compromising fidelity to the source image. In Section 3.1, we describe how such arbitrary deviations can be systematically constrained by regulating the trajectory geometry in latent space. Building upon this foundation, Section 3.2 introduces a custom degraded representation and presents our method to enforce trajectory constraints using this representation to achieve faithful, structure-preserving image edits.

### 3.1 Constraining Editing Trajectories

Consider RF models as the backbone of the editing process, where generative modeling is formulated as learning a transport map between two probability distributions, $\pi_0$ and $\pi_1$. In conventional generative settings, $\pi_0$ denotes the data distribution over images, while $\pi_1$ is a standard Gaussian prior $\mathcal{N}(0, I)$. The RF models parameterize a time-dependent velocity field that induces transporting trajectories between these distributions. The transport is governed by an ODE:

$$dZ_t = v_\theta(Z_t, t)\, dt, \quad Z_0 \sim \pi_0, \; Z_1 \sim \pi_1, \tag{1}$$

where $Z_t$ denotes the state at time $t \in [0, 1]$ and $v_\theta : \mathbb{R}^d \times [0, 1] \to \mathbb{R}^d$ is a time-dependent velocity field driving the flow, parametrized by a neural network. RF assumes a linear interpolation trajectory (straight line) between a data sample $Z_0$ and a noise sample $Z_1$:

$$Z_t = tZ_1 + (1 - t)Z_0. \tag{2}$$

Consequently, the ideal velocity field $v$ that generates this trajectory is constant along the path: $v(Z_t, t) = Z_1 - Z_0$. A neural network $v_\theta(Z_t, t, c)$ is trained to approximate this velocity field, based on condition $c$. To edit a real source image $X_{\text{src}}$, we define two text-conditioned velocity fields: $V^{\text{src}}(Z_t, t) \triangleq v_\theta(Z_t, t, c_{\text{src}})$ corresponding to the source description, and $V^{\text{tar}}(Z_t, t) \triangleq v_\theta(Z_t, t, c_{\text{tar}})$, the target field corresponding to the desired edit. The editing process typically begins by extracting the initial noise map corresponding to the source image, traversing the forward process defined by the ODE:

$$dZ_t^{\text{src}} = V^{\text{src}}(Z_t^{\text{src}}, t) \, dt, \tag{3}$$

where the integration starts at $t = 0$ from the source image $Z_0^{\text{src}} = X_{\text{src}}$ and proceeds to $t = 1$ to reach the noise map $Z_1^{\text{src}}$. Subsequently, the edited image is obtained by solving the reverse ODE using the target velocity field:

$$dZ_t^{\text{tar}} = V^{\text{tar}}(Z_t^{\text{tar}}, t) \, dt. \tag{4}$$

This process is solved backward in time, starting at $t = 1$ with the extracted noise ($Z_1^{\text{tar}} = Z_1^{\text{src}}$) and integrating to $t = 0$ to reach the edited image $Z_0^{\text{tar}}$.

The transition from source to the target distributions via passing through Gaussian noise distribution can also be interpreted as a *direct path* Kulikov et al. (2025) between the source and target distributions. Under this interpretation, instead of transporting samples through the intermediate Gaussian distribution, one can construct a direct path that aligns the forward trajectories of the source and target. A point along this path is defined as:

$$Z_t^{\text{dir}} = Z_0^{\text{src}} + Z_t^{\text{tar}} - Z_t^{\text{src}}. \tag{5}$$

Accordingly, the editing process begins at $t = 1$ with $Z_1^{\text{dir}} = Z_0^{\text{src}}$, and progressively follows the inverse path until reaching $Z_0^{\text{dir}} = Z_0^{\text{tar}}$ at $t = 0$. The corresponding velocity field is then given by

$$V_t^{\text{dir}}(Z_t^{\text{dir}}, t) = V_t^{\text{tar}}(Z_t^{\text{tar}}, t) - V_t^{\text{src}}(Z_t^{\text{src}}, t), \tag{6}$$

which defines the direction of the editing trajectory (see Figure 3a). Importantly, the direct path interpretation extends beyond inverted noise and is equally applicable to cases where $Z_1^{\text{tar}}$ is randomly sampled from Gaussian noise.

The trajectory induced by $V_t^{\text{dir}}$ may potentially take multiple forms, leading to a variety of possible edits. Among these, we seek trajectories that are *constrained* to exclusively modify the intended features dictated by the target prompt. To achieve this, we propose projecting the editing trajectories onto arbitrarily degraded representations where certain image characteristics get suppressed. For instance, the edge representation of an image captures structural information while being blind to intensity and color characteristics. Conversely, an intensely blurred representation preserves overall intensity but is largely invariant to structural details. As illustrated in Figure 2, projecting the editing trajectory onto a degraded representation causes the trajectory to ignore changes to the characteristics that the representation is invariant to.

We can approximate the degraded representation of $V_t^{\text{dir}}$ at each timestep $t$, denoted as $\tilde{V}_t$ as

$$\tilde{V}_t = \mathcal{D}\left(\frac{Z_t^{\text{dir}} - Z_{t-\Delta t}^{\text{dir}}}{\Delta t}\right), \tag{7}$$

where $\mathcal{D}$ is an arbitrary degradation function, and $\Delta t$ is the temporal difference between each step. The projection of $V_t^{\text{dir}}$ on $\tilde{V}_t$ is given by

$$V_t^{\text{proj}} = \frac{\langle V_t^{\text{dir}}, \tilde{V}_t \rangle}{\|\tilde{V}_t\|^2} \tilde{V}_t. \tag{8}$$

Following the projected velocity vector $V_t^{\text{proj}}$ at each timestep $t$ during the editing process ignores changes in the directions that are undefined or suppressed in the corresponding degraded representation. We note

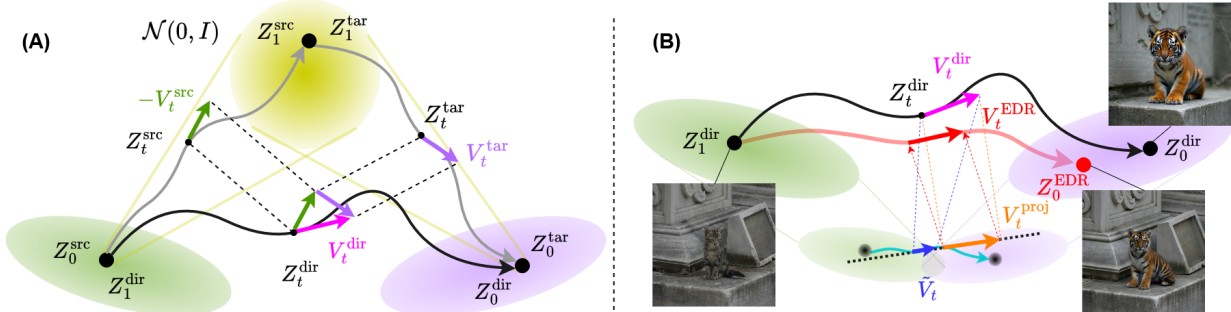

Figure 3: **Proposed EDR Method. (A)** Reinterpretation of the editing process as a direct path, where the velocity $V_t^{\mathrm{dir}}$ is given by the difference between the target and source velocities, $V_t^{\mathrm{tar}} - V_t^{\mathrm{src}}$, as if they were generated directly from the inverted or even random Gaussian noise. **(B)** Our proposed method maps the direct velocity vector $V_t^{\mathrm{dir}}$ to a degraded representation obtained by a combination of Gaussian spatial smoothing and dynamic range reduction. This representation is invariant to minor structural and intensity modifications, which are commonly caused by secondary editing effects. The initial update velocity vector is then projected onto this degraded counterpart to suppress these unintended modifications, resulting in more precise and focused edits.

that the edge representation serves purely as an illustration of this principle; the deployed method (Section 3.2) relies only on Gaussian structural suppression and dynamic range reduction. While Figure 2 depicts an idealized projection onto a subspace of retained characteristics, deriving such a projection operator for the highly non-linear characteristics of generative latents is intractable at inference time; the single-direction projection of Equation 8 therefore acts as a computationally efficient surrogate.

## 3.2 Editing through Degraded Representations (EDR)

Here, we introduce our method to leverage the capability of restraining editing trajectories along undesired directions, ensuring that the edits remain precise and faithful to the source image. We categorize the modifications occurring during the editing process into two main types: primary modifications, which implement the intended edit and are often substantial, and secondary modifications, which constitute minor side effects. For example, consider editing an image of a blue car into a red car. In an ideal editing process, the color is successfully changed as intended (primary modification), but the car shape may inadvertently become more *sporty* due to spurious correlation between red color and sports cars, representing a secondary modification.

In principle, the primary modification represents the main focus of the model, ensuring that the target image aligns with the prompt text embedding, as opposed to secondary modifications, which constitute minor side effects that are not explicitly guided by the prompt. Hence, we need to project the editing trajectory onto a representation space that is invariant to subtle, insignificant modifications which form the secondary modifications in our context.

We characterize modifications into structural and color distribution shifts, allowing us to target each specifically through our degraded representation. To design a representation that is invariant to insignificant amount of these two modifications, we use Gaussian spatial smoothing and dynamic range reduction. Formally, for $Z_t^{\mathrm{dir}}$, our proposed combined degraded representation is

$$\mathcal{D}_{\mathrm{c}}(Z_t^{\mathrm{dir}}) = I_{\min} + (I_{\max} - I_{\min})\left(G_\sigma * Z_t^{\mathrm{dir}}\right), \quad 0 < I_{\min} < I_{\max} < 1, \tag{9}$$

where $G_\sigma$ is the Gaussian kernel with standard deviation $\sigma$ and the size equal to $6\sigma$, while $I_{\min}$ and $I_{\max}$ define the target intensity range. In our proposed method, we use $\mathcal{D}_{\mathrm{c}}(\cdot)$ in place of $\mathcal{D}(.)$ in Equation 7.

Projecting $V_t^{\mathrm{dir}}$ onto this degraded representation constrains the editing trajectory against incidental structural and intensity shifts, ensuring the transformation remains aligned with the source image's foundational characteristics. Importantly, this mechanism does not strictly confine the model to the source state; since the

designed degraded representation $\mathcal{D}_c$ remains sensitive to substantial structural and color transformations, it permits significant semantic changes when necessitated by the target prompt.

While projecting the trajectory onto our proposed degraded representation effectively enforces fidelity to the source, it potentially can compromise the quality of target image. Therefore, we utilize the projected trajectory in a decaying manner, to ensure preservation of the image quality. The update vector of the proposed method, denoted as $V_t^{\text{EDR}}$, is given by

$$V_t^{\text{EDR}} = \alpha V_t^{\text{proj}} + (1 - \alpha)V_t^{\text{dir}}. \tag{10}$$

Here, the coefficient $\alpha$ scales the contribution of the projected trajectory according to the decay rate $\gamma$ as

$$\alpha = t^\gamma. \tag{11}$$

Furthermore, the overall extent of the modification is governed by the editing strength, denoted as $t_0$, which dictates the starting timestep of the generative process. A larger $t_0$ provides the model with a longer trajectory to satisfy complex target prompts, whereas a smaller $t_0$ inherently restricts the generation closer to the source image.

Thus, starting from $t_0$, our method prioritizes the projected update vector $V_t^{\text{proj}}$ during the initial timesteps to guide the edit along the degraded representation. This ensures structural fidelity to the source image without deviating into irrelevant directions. As the generation progresses into the later steps, the update vector $V_t^{\text{EDR}}$ gradually shifts its weight toward the model's native directional vector $V_t^{\text{dir}}$, ensuring the final output is realistic and visually coherent. Figure 3b illustrates the proposed method with $\gamma = 0$ where $V_t^{\text{EDR}}$ is updated only with regard to $V_t^{\text{proj}}$.

## 4 Experimental Evaluation

### 4.1 Setup

#### 4.1.1 Experimental Setting

We evaluate our approach on two state-of-the-art RF-based text-to-image models: Stable Diffusion 3 (SD3) Esser et al. (2024) and FLUX.1-dev Labs et al. (2025). For SD3, we set the total inference steps to T=50. The editing strength ($t_0$) is set to 0.76 with a decay factor $\gamma = 2$. We employ a Classifier-Free Guidance (CFG) scale of 3.5 and 13.5 for source and target images, respectively. For FLUX.1-dev, we utilize $T = 28$ inference steps. The corresponding editing parameters are configured as $t_0 = 0.9$ and $\gamma = 5$. For both models, Gaussian smoothing is applied using $\sigma = 5$ and a kernel size of 30. The reduced dynamic range is bounded between $I_{min} = 0.25$ and $I_{max} = 0.75$. We chose these hyperparameters empirically (see supplementary material).

#### 4.1.2 Comparison Methods

We evaluate our approach against the following methods: the baseline SDEdit Meng et al. (2021), which applies editing by adding random Gaussian noise followed by denoising conditioned on target prompt, ODE Inversion as discussed in Section 3.1, and state-of-the-art flow-based methods RF inversion Rout et al. (2024), RF Edit Wang et al. (2024), iRFDS Yang et al. (2024), and FlowEdit Kulikov et al. (2025).

For the SDEdit baseline, we set the editing strength to $t_0 = 0.4$ with a target Classifier-Free Guidance (CFG) scale of 13.5 on SD3. For FLUX, we utilized $t_0 = 0.75$ with a target CFG scale of 5.5. For RF Edit, we used its official implementation which is applied only on FLUX and used default hyperparameters: 25 guidance steps, 5 injection steps, and a guidance scale of 3.5. For FlowEdit, we utilized the provided codebase and the hyperparameter settings recommended in the original paper Kulikov et al. (2025). For SD3, we set the total number of steps to $T = 50$, with the editing starting timestep at $n_{\max} = 33$, and CFG scales of 3.5 and 13.5 for the source and target conditioning, respectively. For FLUX, the parameters were set to $T = 28$ steps, $n_{\max} = 24$, and CFG scales of 1.5 and 5.5 for the source and target. Also, for ODE inversion we applied the same hyperparameters.

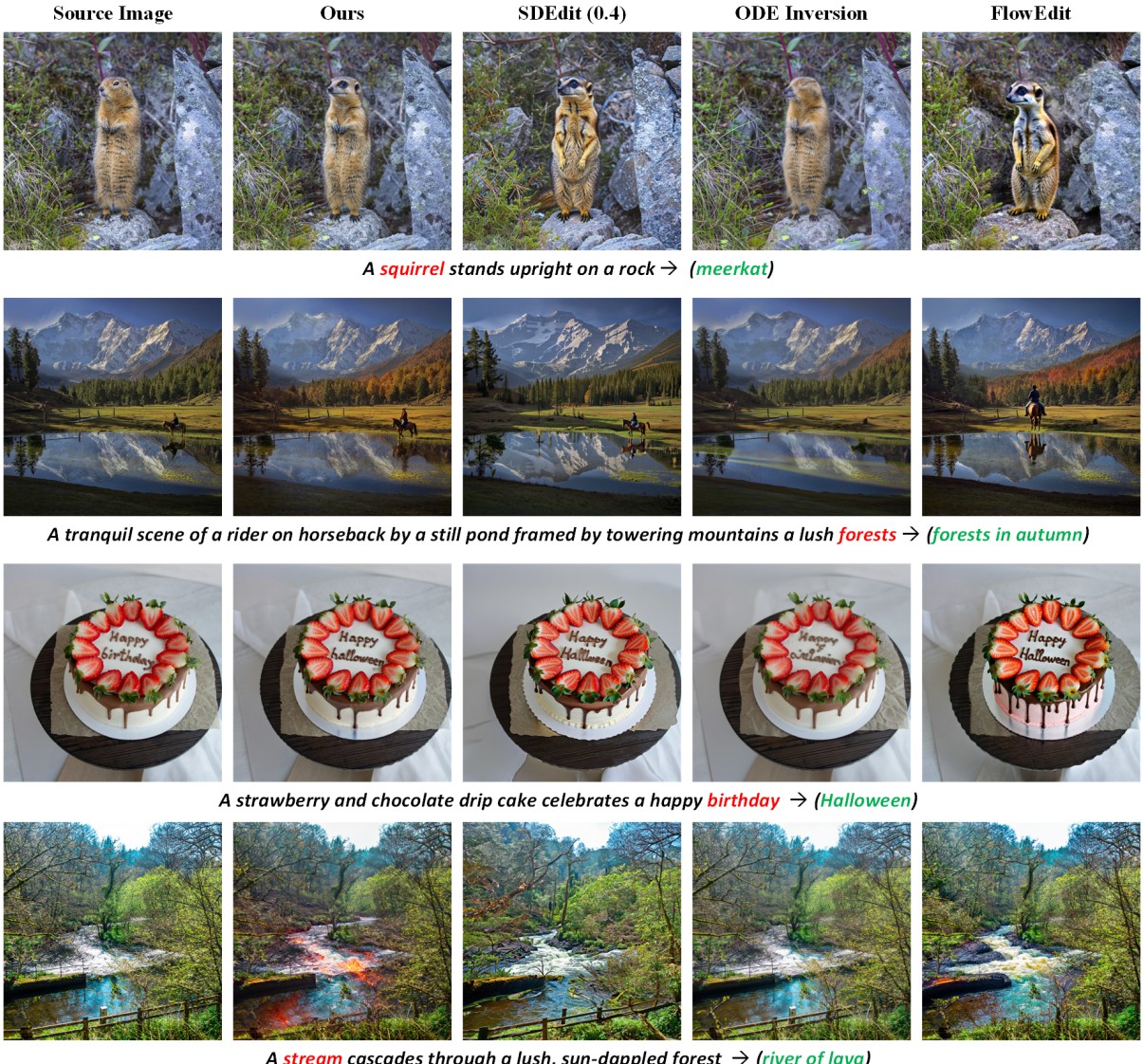

| Source Image | Ours | SDEdit (0.4) | ODE Inversion | FlowEdit |

*A **squirrel** stands upright on a rock → **(meerkat)***

*A tranquil scene of a rider on horseback by a still pond framed by towering mountains a lush **forests** → **(forests in autumn)***

*A strawberry and chocolate drip cake celebrates a happy **birthday** → **(Halloween)***

*A **stream** cascades through a lush, sun-dappled forest → **(river of lava)***

Figure 4: **Qualitative results on SD3.** Our method not only achieves higher fidelity to the source image but also facilitates more effective semantic editing.

### 4.1.3 Dataset

For qualitative comparisons, we curated a diverse evaluation set comprising 80 high-resolution images. This dataset is sourced from LSDIR Li et al. (2023), DIV2K Agustsson & Timofte (2017), and the dataset introduced by Kulikov *et al.* in Kulikov et al. (2025). To ensure accurate image-to-text alignment, we utilize Gemma-3-12B Gemma Team & Google DeepMind (2025) to infer descriptive source prompts for each image. Subsequently, for every image-prompt pair, we manually craft two distinct target prompts to evaluate various editing scenarios, resulting in a total of 160 test cases. The quantitative evaluation of our method is executed on the dataset and prompts in Kulikov et al. (2025) to ensure a fair comparison.

### 4.2 Qualitative Evaluation

The most reliable evaluation of image editing quality is arguably qualitative assessment by human users. Therefore, we first evaluate our method qualitatively by performing a user study. We chose four metrics

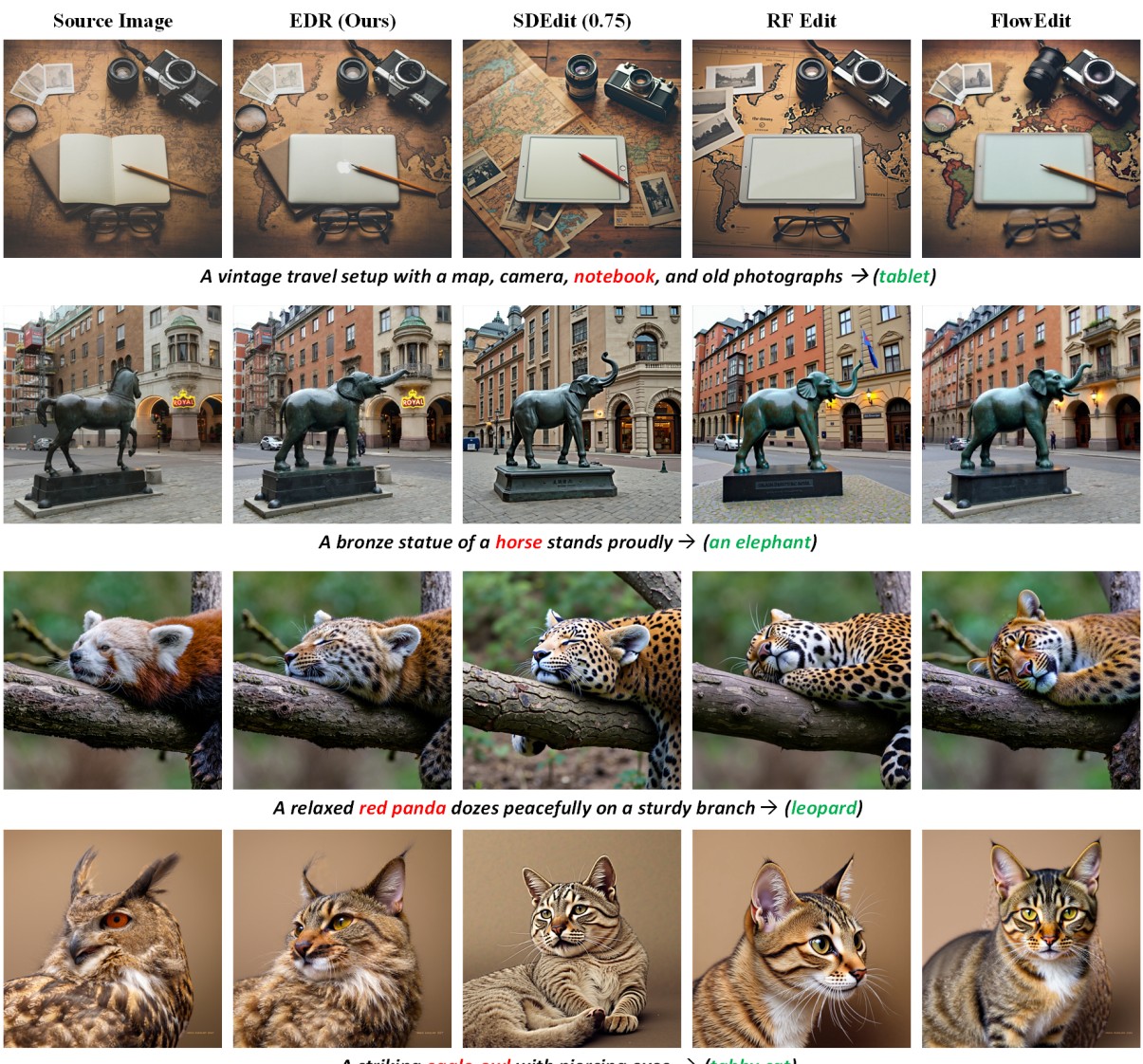

Figure 5: **Qualitative results on FLUX.** Our method produces high-quality edits while effectively preserving the background and subject pose of the source image.

for this evaluation, namely Prompt Adherence, Scene Preservation, Pose Preservation, and overall Success. Prompt Adherence reflects the degree to which the edited image correlated with the given target prompt. The structural faithfulness is divided into two subcategories: Scene Preservation and Pose Preservation. Eventually, Overall Success gives the unified success of the editing. We performed the user study on 30 participants, each asked to compare the edited images on the four criteria given each source image and target prompt (see Appendix B for the complete protocol and inter-rater agreement). We report the win rate of our method against compared methods on SD3 in Table 1 and on FLUX in Table 2. Representative results of our method using SD3 and FLUX are given in Figure 4 and Figure 5, respectively.

## 4.3 Quantitative Evaluation

To quantitatively evaluate our proposed method, we employ the following metrics: cosine similarity of CLIP Radford et al. (2021) embeddings—specifically $CLIP_{txt}$ to measure the alignment between the target image and the editing prompt (quantifying editing success), and $CLIP_{img}$ to assess the similarity between the

Table 1: **User study on SD3.** The win rate (%) of our method against competing approaches across four criteria is presented. Our method achieves superior performance in editing effectiveness (Prompt Adherence), source image fidelity (Scene and Pose Preservation), and Overall Success compared to existing methods on SD3.

|                    | SDEdit | ODE Inv. | FlowEdit |
|--------------------|--------|----------|----------|
| Prompt Adherence   | 79 %   | 96 %     | 72 %     |
| Scene Preservation | 83 %   | 74 %     | 67 %     |
| Pose Preservation  | 77 %   | 75 %     | 78 %     |
| Overall Success    | 81 %   | 93 %     | 75 %     |

Table 2: **User study on FLUX.** We report the win rate (%) of our method against competing approaches. Our method remains significantly more faithful to the source images while maintaining prompt adherence on par with state-of-the-art baselines.

|                    | SDEdit | RF Edit | FlowEdit |
|--------------------|--------|---------|----------|
| Prompt Adherence   | 57 %   | 52 %    | 50 %     |
| Scene Preservation | 94 %   | 71 %    | 67 %     |
| Pose Preservation  | 82 %   | 89 %    | 85 %     |
| Overall Success    | 89 %   | 83 %    | 78 %     |

source and target images. Additionally, we report DINO Caron et al. (2021) embedding similarity to further evaluate the structural preservation between the source and target images. Also, we used LPIPS Zhang et al. (2018) to measure the perceptual distance between source and target images, and DreamSim Fu et al. (2023) that assesses object pose and holistic perceptual similarities in source and target images. In our quantitative evaluation, we adopted the dataset and the results for comparison methods from Kulikov et al. (2025). Quantitative results on SD3 and FLUX are given in Table 3 and Table 4, respectively. More results are given in the supplementary material.

## 5 Ablation Study

The proposed approach utilized degraded representation that is a combination of two different degradations, namely Gaussian structural suppression (GSS) and dynamic range reduction (DRR). To verify that both these degradations are essential and complementary for the task, we executed experiments with dropping each of the operators. Figure 6 shows the effect of ablation of the two degradation operators. As expected, without dynamic range reduction the color palette of the edited image changed significantly. On the other hand, ablating Gaussian blur degradation, resulted in an edited image that drifted from the source image structure.

Furthermore, Table 5 compares our proposed degradation representation against baselines that omit individual components. Removing any single operator severely compromises overall fidelity. While ablating GSS and DRR yields a slight increase in editing strength $\text{CLIP}_{\text{txt}}$, this inflated text alignment score occurs because the generation trajectories become completely unconstrained. In this case, the model easily satisfies the target prompt only by ignoring structural preservation, which is reflected in the severe degradation across all fidelity metrics $\text{CLIP}_{\text{img}}$, DINO, LPIPS, and DreamSim.

Table 3: **Quantitative results on SD3.** Our proposed method achieves state-of-the-art performance, particularly in metrics evaluating structural preservation. The best values are highlighted in **bold**.

|                            | $\text{CLIP}_{\text{txt}}\uparrow$ | $\text{CLIP}_{\text{img}}\uparrow$ | DINO $\uparrow$ | LPIPS $\downarrow$ | DreamSim $\downarrow$ |
|----------------------------|----------|----------|----------|----------|----------|
| SDEditMeng et al. (2021)   | 0.330    | 0.885    | 0.634    | 0.251    | 0.213    |
| iRFDSYang et al. (2024)    | 0.335    | 0.822    | 0.534    | 0.376    | 0.327    |
| FlowEditKulikov et al. (2025) | 0.344 | 0.872    | 0.719    | 0.181    | 0.253    |
| EDR (Ours)                 | **0.347** | **0.909** | **0.744** | **0.146** | **0.158** |

Table 4: **Quantitative results on FLUX.** Our proposed method achieves state-of-the-art performance on metrics evaluating structural fidelity, while maintaining effective editing capabilities. The best values are highlighted in **bold**.

| | CLIP$_{txt}$↑ | CLIP$_{img}$↑ | DINO ↑ | LPIPS ↓ | DreamSim ↓ |
|---|---|---|---|---|---|
| SDEdit Meng et al. (2021) | 0.316 | 0.902 | 0.637 | 0.264 | 0.180 |
| RF Inversion Rout et al. (2024) | 0.334 | 0.856 | 0.558 | 0.34 | 0.266 |
| RF Edit Wang et al. (2024) | 0.332 | 0.876 | 0.650 | 0.220 | 0.220 |
| FlowEdit Kulikov et al. (2025) | **0.337** | 0.875 | 0.682 | 0.223 | 0.252 |
| EDR (Ours) | 0.335 | **0.914** | **0.739** | **0.170** | **0.136** |

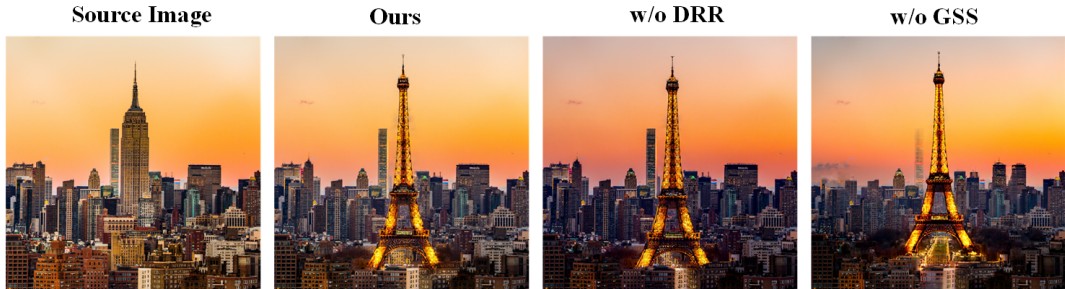

Figure 6: **Ablation Impact** Ablating dynamic range reduction (DRR) and Gaussian structural suppression (GSS) from our proposed degraded representation. Omitting DRR leads to an editing that is not faithful to the original image intensity and color palette, and omitting GSS produces an edited image unfaithful to the source structure.

# 6 Conclusion

In this work, we addressed the challenge of achieving faithful and effective image editing utilizing flow-based models. We demonstrated that the generation trajectories between two distinct distributions can be systematically restricted from moving in arbitrary directions. This is accomplished by projecting the trajectories onto degraded representations that are inherently invariant to such changes. Consequently, we proposed a method of constraining editing trajectories to prevent unintended visual deviations by leveraging these degraded representations, wherein specific image properties, namely pixel intensities and overall structure, are suppressed.

## Limitations

EDR is designed for faithful editing, and its principal limitation is the boundary of that objective: drastic semantic edits in which the source and target share little structural correspondence. When the target prompt requires large changes in shape, category, or layout — for example, transforming a cat into an

Table 5: **Ablation Study.** Dropping either of the degradation operators, GSS or DRR, results in a significant drop in metrics that measure editing fidelity.

(a) Ablation Study on SD3

| | CLIP$_{txt}$↑ | CLIP$_{img}$↑ | DINO ↑ | LPIPS ↓ | DreamSim ↓ |
|---|---|---|---|---|---|
| w/o GSS | 0.347 | 0.873 | 0.612 | 0.287 | 0.278 |
| w/o DRR | 0.349 | 0.865 | 0.604 | 0.233 | 0.291 |
| EDR | 0.347 | 0.909 | 0.744 | 0.146 | 0.158 |

(b) Ablation Study on FLUX.1-dev

| | CLIP$_{txt}$↑ | CLIP$_{img}$↑ | DINO ↑ | LPIPS ↓ | DreamSim ↓ |
|---|---|---|---|---|---|
| w/o GSS | 0.336 | 0.847 | 0.582 | 0.291 | 0.225 |
| w/o DRR | 0.340 | 0.822 | 0.556 | 0.258 | 0.239 |
| EDR | 0.335 | 0.914 | 0.739 | 0.170 | 0.136 |

elephant, or replacing an object with one of a substantially different geometry — the trajectory constraint, which suppresses deviations from the source structure most strongly in the early editing steps, opposes the very modifications the prompt demands. In such cases the method resolves the conflict toward fidelity: the characteristic failure mode is an under-edited output that remains close to the source rather than an unfaithful or corrupted one. Relatedly, since the degraded representation is built from spatial smoothing, edits whose primary content is fine-grained and spatially localized in the early trajectory can require a larger editing strength $t_0$ to fully materialize. We regard this behavior as the intended operating point of the fidelity–editability trade-off — faithful editing is well-posed only when a meaningful structural relation between source and target exists — but it delimits the scope of EDR: for creative transformations that deliberately discard source structure, unconstrained editors or pure generation are the more suitable tools. Extending the degraded representation to adapt its invariances to the magnitude of the requested edit is an interesting direction for future work.

## Broader Impact Statement

EDR enables high-fidelity, mask-free, prompt-driven editing of real photographs while preserving the structure and identity of the source image. Alongside beneficial applications in creative content production, photo restoration, and design, this capability carries dual-use risks: photorealistic manipulation of real images can facilitate misinformation, non-consensual edits of identifiable individuals, and deepfake-style content, and the strong source fidelity of our edits may make such manipulations harder to identify by visual inspection. We note that EDR builds entirely on publicly available pre-trained rectified flow backbones (SD3 and FLUX.1-dev) and introduces no new generative capability: it constrains editing trajectories of models already broadly accessible through existing editors, restricting what an edit may alter rather than enlarging the space of what can be synthesized, so the risk it adds is incremental over already-available editors. As mitigations, our implementation embeds an invisible blind watermark in EDR outputs by default, serving as a provenance signal for good-faith use, and our released code will be accompanied by usage guidelines requiring compliance with the acceptable-use policies of the underlying models, which prohibit deceptive manipulation of real individuals. Moreover, because EDR outputs are decoded through the same pre-trained autoencoders as standard SD3 and FLUX generations, existing synthetic-image detectors and provenance mechanisms developed for these models (e.g., content-credential standards such as C2PA) remain applicable. This work motivates continued research on detecting AI-edited imagery.

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

# Appendix

## A    Additional Results

Additional results on SD3 and FLUX using the proposed method EDR are presented in Figure A.

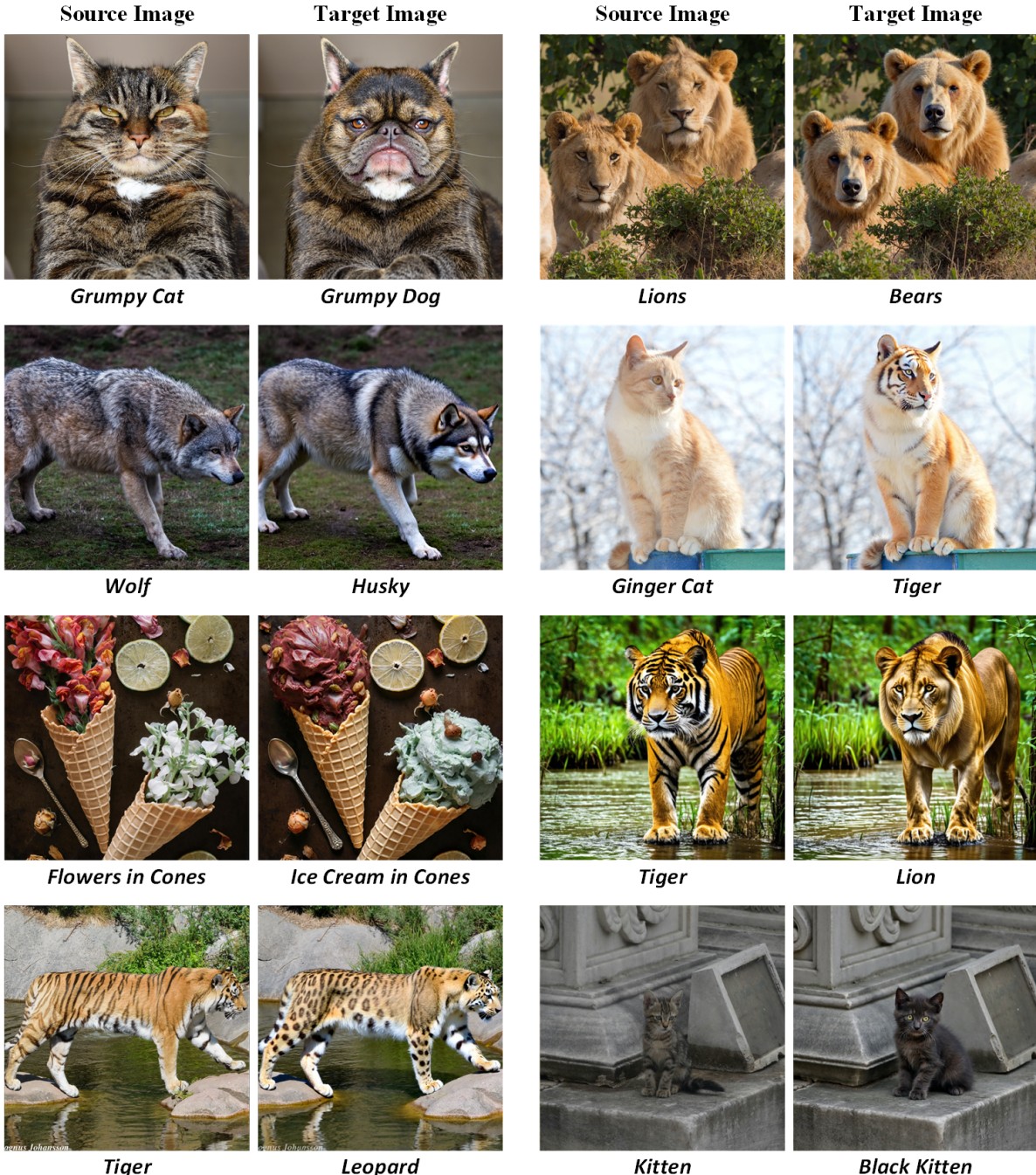

Figure A: Additional results of EDR and FLUX (top 2 rows) and SD3 (bottom 2 rows)

**... in Pixar Style**

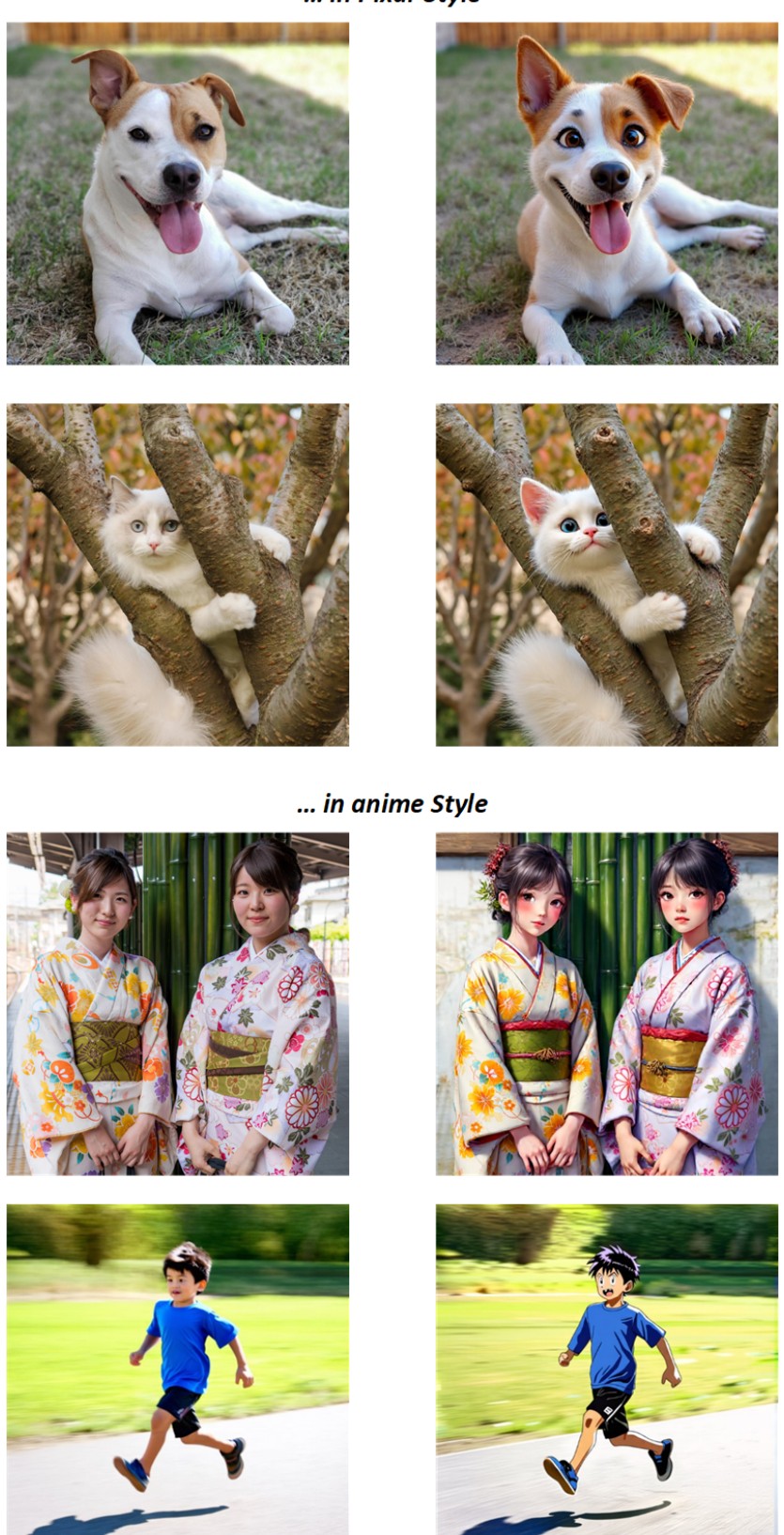

**... in anime Style**

Figure B: EDR for style editing

Furthermore, our method is highly effective and faithful for complex style editing tasks (Figure B), successfully adopting target aesthetics while strictly preserving the source image's structural integrity.

## B    User Study Details

We provide here the complete protocol of the user study reported in Section 4.2.

**Protocol.**   The study covers 160 test cases (80 images $\times$ 2 target prompts) per backbone (SD3 and FLUX.1-dev). For each test case, the output of our method is compared against the output of each comparison method in an independent pairwise trial, rather than a simultaneous ranking of all methods; we adopted this design because all methods share the same backbone and can produce very similar results for a given case, in which situation a simultaneous ranking of four outputs would be largely arbitrary for the similar ones. In each trial, participants are shown the source image, the target prompt, and the two anonymized outputs in randomized order, and indicate the superior output for each of the four criteria (Prompt Adherence, Scene Preservation, Pose Preservation, and Overall Success). The choice is a forced binary decision with no tie option, so ties cannot occur. Each pairwise trial is judged by three different participants to ensure robustness to individual preferences. Per backbone, this amounts to 480 pairwise trials (160 test cases $\times$ 3 comparison methods); distributed among the 30 participants.

**Win-rate computation.**   The win rates reported in Tables 1 and 2 are the percentage of individual judgments in which our method is preferred over the corresponding comparison method.

**Inter-rater agreement.**   To assess the reliability of the judgments, we report the mean pairwise raw agreement: for each trial and criterion, the fraction of the three rater pairs giving the same preference, averaged over all trials, reported per criterion and per comparison method in Tables A and B. Agreement ranges from 0.74 to 0.97 across criteria and comparison methods, well above the 0.5 chance level of a binary preference, indicating substantial consistency among raters.

Table A: Mean pairwise raw agreement of the user study on SD3

|                    | SDEdit | ODE Inv. | FlowEdit |
|--------------------|--------|----------|----------|
| Prompt Adherence   | 0.8542 | 0.8583   | 0.8250   |
| Scene Preservation | 0.9333 | 0.8958   | 0.8333   |
| Pose Preservation  | 0.9000 | 0.8917   | 0.8458   |
| Overall Success    | 0.8292 | 0.8500   | 0.8625   |

Table B: Mean pairwise raw agreement of the user study on FLUX.1-dev

|                    | SDEdit | RF Edit | FlowEdit |
|--------------------|--------|---------|----------|
| Prompt Adherence   | 0.8083 | 0.8208  | 0.7417   |
| Scene Preservation | 0.9708 | 0.8625  | 0.7875   |
| Pose Preservation  | 0.8792 | 0.9167  | 0.8833   |
| Overall Success    | 0.8708 | 0.8125  | 0.8542   |

## C    Efficiency Study

Since avoiding per-image optimization and multi-prediction aggregation is a stated motivation of our method, we measure its computational cost directly. We report wall-clock time per edit and peak GPU memory for our method and the comparison methods on the first five edits of the quantitative dataset, using the experimental settings of Section 4.1. Results are given in Table C for SD3 and Table D for FLUX (mean $\pm$ standard deviation over the five edits).

EDR requires less runtime than FlowEdit on both backbones and than RF Edit on FLUX, with equal or lower peak memory in all cases, consistent with our method requiring no aggregation of multiple model predictions per step. The added operations of EDR — a separable Gaussian smoothing and a vector projection — introduce no additional network evaluations and are orders of magnitude cheaper than a single backbone forward pass. Among the compared methods, ODE inversion attains the lowest runtime on SD3, but at a substantial fidelity cost, as reflected in the user study (Table 1), where our method achieves a 93% overall-success win rate against it.

We additionally benchmark Imagic Kawar et al. (2023) as a representative optimization-based method, using a publicly available implementation built on Stable Diffusion v1.5 (Table E). Despite the relatively small backbone, it requires almost five minutes per image, confirming that per-image optimization is not competitive with optimization-free methods in terms of time efficiency.

Table C: Efficiency results on SD3

|  | EDR (Ours) | FlowEdit ($n_{avg}$=2) | ODE Inversion |
| --- | --- | --- | --- |
| Wall-clock time (s) | $13.60 \pm 0.31$ | $18.25 \pm 1.26$ | $6.47 \pm 0.33$ |
| Peak memory (GB) | 19 | 19 | 19 |

Table D: Efficiency results on FLUX.1-dev

|  | EDR (Ours) | FlowEdit ($n_{avg}$=2) | RF Edit |
| --- | --- | --- | --- |
| Wall-clock time (s) | $23.94 \pm 0.63$ | $44.59 \pm 0.24$ | $55.99 \pm 0.12$ |
| Peak memory (GB) | 33 | 33 | 35 |

Table E: Efficiency results for the optimization-based method Imagic Kawar et al. (2023) on Stable Diffusion v1.5

|  | Imagic |
| --- | --- |
| Wall-clock time (s) | $299.4 \pm 0.8$ |
| Peak memory (GB) | 17 |

## D   Hyperparameter Selection

The hyperparameters used in our method were selected based on a comprehensive set of experiments. Specifically, we executed a grid search over the standard deviation $\sigma$ (used for Gaussian structural smoothing) and the reduced dynamic range $[I_{min}, I_{max}]$ on SD3. We evaluated these combinations by plotting $\text{CLIP}_{txt}$ to measure editing success against LPIPS to measure source image fidelity. As shown in Figure C, the tested combinations form a distinct trade-off curve. We select the configuration $\sigma = 5$ and $[I_{min}, I_{max}] = [0.25, 0.75]$ (green triangle), as it sits on the Pareto front and achieves the best empirical balance without compromising either metric.

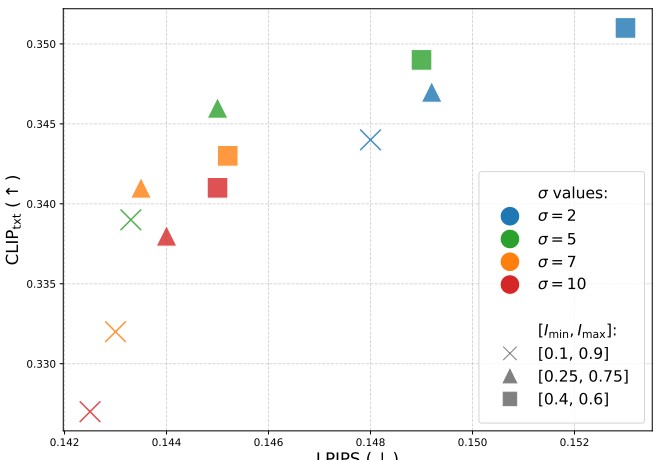

Figure C: **Hyperparameter Grid Search.** Quantitative evaluation of different configurations for the structural smoothing factor $\sigma$ and the intervention bounds $[I_{\min}, I_{\max}]$. The scatter plot illustrates the trade-off between editability (measured by $\text{CLIP}_{\text{txt}}$, where higher is better) and source image fidelity (measured by LPIPS, where lower is better). The green triangle denotes our selected optimal configuration ($\sigma = 5$ and $[I_{\min}, I_{\max}] = [0.25, 0.75]$) situated on the Pareto front, achieving the best balance between successful text-driven editing and structural preservation.

## E    Pseudocode

The complete pseudocode of the proposed method (EDR) is presented in Algorithm A.

---

**Algorithm A:** Editing via Degraded Representations (EDR)

---

**Input:** Source image $X_{\text{src}}$, source and target prompts $c_{\text{src}}$ and $c_{\text{tar}}$, degradation map $\mathcal{D}_c(.)$ (standard deviation $\sigma$, reduced dynamic range $[I_{\min}, I_{\max}]$) based on Eq. 9 in the main paper, total timesteps $T$, editing strength $t_0$, and decay rate $\gamma$

**Output:** Edited image $X_{\text{tar}}$

---

**1** $n \leftarrow \lfloor t_0 \times T \rfloor$

**2** $t \leftarrow t_0$

**3** $Z_t^{\text{dir}} \leftarrow X_{\text{src}}$

**4** $\Delta t \leftarrow \frac{t_0}{n}$

**5** **for** $i = n$ **to** $1$ **do**

**6** $\quad N \sim (0, I)$

**7** $\quad Z_t^{\text{src}} \leftarrow (1-t)Z_t + tN$

**8** $\quad Z_t^{\text{tar}} \leftarrow Z_t^{\text{dir}} + Z_t^{\text{src}} - Z_0^{\text{src}}$

**9** $\quad V_t^{\text{dir}}(Z_t^{\text{dir}}, t) \leftarrow V_t^{\text{tar}}(Z_t^{\text{tar}}, t) - V_t^{\text{src}}(Z_t^{\text{src}}, t)$

**10** $\quad \tilde{V}_t = \mathcal{D}_c\left(\frac{\Delta Z_t^{\text{dir}}}{\Delta t}\right)$

**11** $\quad V_t^{\text{proj}} = \frac{\langle V_t^{\text{dir}}, \tilde{V}_t \rangle}{\|\tilde{V}_t\|^2} \tilde{V}_t$

**12** $\quad \alpha = t^\gamma$

**13** $\quad V_t^{\text{EDR}} = \alpha V_t^{\text{proj}} + (1-\alpha)V_t^{\text{dir}}$

**14** $\quad Z_{t-\Delta t}^{\text{dir}} \leftarrow Z_t^{\text{dir}} + V_t^{\text{EDR}}$

**15** $\quad t \leftarrow t - \Delta t$

**16** **return** $X_{\text{tar}} = Z_t^{\text{dir}}$

---

# F    Watermarking of EDR Outputs

As stated in the Broader Impact statement, our implementation embeds an invisible blind watermark in EDR outputs by default. We use the attention-based watermarking method of Zhang et al. (2019), applied in pixel space to the decoded output image with a 32-bit payload. The embedding is enabled by default in our implementation and must be explicitly disabled.

**Imperceptibility.**    We measure the perceptual impact of the embedding on the EDR outputs of our evaluation set by comparing each watermarked image against its unwatermarked counterpart. The embedding is imperceptible, with a mean PSNR of 40.66 dB and a mean SSIM of 0.973. The quantitative results in Section 4.3 are computed on outputs prior to embedding; given the measurements above, the effect of the embedding on these metrics is negligible.

**Robustness.**    We verify payload recovery under common post-processing on the watermarked EDR outputs of our evaluation set. Recovery is near-exact across all tested perturbations: the mean bit accuracy is 99.2% with no perturbation, 99.1% under Gaussian blurring ($\sigma = 1$), 99.1% under $0.5\times$ downscaling, and 98.1% under JPEG compression (quality 80). This high robustness at mild perturbation strengths is consistent with the adversarial robustness training of Zhang et al. (2019).

**Scope.**    We regard this watermark as a provenance signal for good-faith use rather than a defense against a determined adversary, who could disable the embedding. Complementary provenance mechanisms such as content-credential standards (e.g., C2PA) apply to EDR outputs as to standard SD3 and FLUX generations.

