# OpenReview forum: "Faithful Image Editing via Degraded Representations"
_TMLR — Under review for TMLR_

### Review · Reviewer_VVF8 · 2026-06-07

**Summary Of Contributions:**

The paper proposes a method for image generation/editing/synthesis based on Rectified Flow concept. The core contribution lies in  a mechanism for constraining generative editing trajectories. The method is evaluated both numerically as well as subjectively and compared to recent and relevant proposals.

**Additional Comments:**

Overall, I view the paper as follows:

1. On the technical side, the paper tends to be marginally above the acceptance threshold (conditioned on the details of the subjective evaluation).
     -    The technical contribution exists, but given the prior work on receptive fields, it is more incremental in nature.
     -    The results: the objective metrics are compelling. However, the subjective evaluation report is missing important details. The images shown look good, but not exceptional.
2. There is a critical issue regarding ethical concerns and potential misuse.

**Audience:**

Yes

**Audience Explanation:**

Actually, in the context of this paper, this is not a good question ("if there are people interested"); thus, directly answering it is not very relevant.

The key problem of this paper is the ethical concern described below. Given that I do have concerns about the possible misuses of the proposed solution, it is not relevant whether there are people interested in the findings. I argue that a more relevant question in this context is: "Are there interested individuals who aim to use the findings in this paper in a positive way and, *at the same time*, can we isolate individuals who may use the findings in a harmful way?"

I believe that well-intentioned individuals will be interested in the findings, but unfortunately, the paper provides no solution to isolate or safeguard against misuse.

**Broader Impact Concerns:**

Unfortunately, the paper is significantly lacking in this regard. The core contribution is a method for editing images guided by a given prompt, with the goal of producing more realistic outputs.

However, we are no longer in the early stages of AI development, where it was reasonable to assume that all technological advances are to be used solely for beneficial purposes. A substantial body of literature has demonstrated that  image editing can be misused in a variety of harmful ways. The paper completely ignores these concerns.

Given the many potential negative applications of image generation technologies, I believe that new contributions in this area should not be published without not only a thorough discussion of the associated risks, but also a **rigorous experimental evaluation of safety aspects**. For example, the paper should assess whether images edited by the proposed method can still be reliably detected as synthetic and investigate other mechanisms that could help establish safety safeguards and limit misuse.

**Claims And Evidence:**

No

**Claims Explanation:**

With respect to the claims made (stated on page 3), I believe that the paper provides some evidence, but important details are missing from the experimental section (particularly regarding Claim 3).

The main criticism I have regarding the claims is the statement "maintaining exceptional source fidelity." *Exceptional*. The paper reports results on only 80 high-resolution images. Regardless of the numerical results, 80 images are not sufficient to support a claim of "exceptional" performance.

Furthermore, several aspects are insufficiently detailed:

1. The paper does not explain what is actually meant by *source fidelity*. To make myself clear, let us focus on Figure 4, first example: ground squirrel vs. meerkat. Why is switching only the head of the meerkat onto the squirrel's body considered the desired effect, rather than replacing the squirrel entirely with a fully recognizable meerkat?
While the paper is limited to 12 pages, some explanation could be added to the main paper, with additional details provided in the appendix.
2. Section 4.2 only vaguely describes the experiment that led to Tables 1 and 2. I am guessing that four sets of experiments were conducted, and pairs of images (proposed-edited vs. SOTA-edited) were presented to users, who were then asked to answer four questions. If this understanding is correct, the following issues arise:
    -   What is the agreement between users on the same set of data? As currently presented, it appears that each image pair was labeled by a single user, which is not very informative.
    -   Why did the experiment not involve all images and all methods, allowing a ranking according to each criterion to be reported? The current setup may allow cherry-picking of certain image pairs or scenes.

**Requested Changes:**

1. Add explanation w.r.t subjective evaluation
2. Mandatory - add discussion and evaluation if the proposed method may be safeguarded against misuse .

---

> ### Author Response · Authors · 2026-07-05
> **Response to Reviewer VVF8 (1/3)**
>
> We thank the reviewer for the thoughtful and thorough review. We respond to each concern point by point below, and the corresponding changes have been incorporated into the revised manuscript.
>
> **Claims:**
> We understand that qualitative evaluation is inherently subjective, and that the term "exceptional" may evoke excessive expectations. Therefore, we replaced "exceptional source fidelity'' with "remarkable source fidelity".
>
>
> **Insufficiently detailed aspects:**
>
> **1.**
> Thank you for this constructive feedback. The "source fidelity" mainly refers to "pose" and "scene (background)" preservation as reflected in our qualitative evaluation. We will explicitly mention this in the paper.
>
> Regarding the "squirrel" example, it is exactly what we mean by "faithful editing", as our result provides "scene preservation" and "pose preservation". The result would be more natural if the squirrel were replaced completely by a "meerkat", but in this work, our focus is on "faithful editing", which we achieved by restricting the editing trajectory from deviation in unwanted directions to preserve the overall shape and intensity.
>
> In the revision, we define source fidelity at first use as preservation of scene composition, background, subject pose, and global color statistics, with modifications restricted to those entailed by the target prompt, with the squirrel–meerkat case discussed in the appendix as an illustration of this operating point.
>
> **2.**
> Each pair is evaluated by three different participants, and we
> report the percentage of individual judgments in which our method
> is preferred over each comparison method. The mean pairwise raw
> agreement --- for each trial and criterion, the fraction of the
> three rater pairs giving the same preference, averaged over all
> trials --- is given per criterion and per baseline in the tables
> below.
>
> Mean Pairwise Raw Agreement for SD3:
>
> |                    | SDEdit | ODE Inv. | FlowEdit |
> |--------------------|--------|----------|----------|
> | Prompt Adherence   | 0.8542 | 0.8583   | 0.8250   |
> | Scene Preservation | 0.9333 | 0.8958   | 0.8333   |
> | Pose Preservation  | 0.9000 | 0.8917   | 0.8458   |
> | Overall Success    | 0.8292 | 0.8500   | 0.8625   |
>
> Mean Pairwise Raw Agreement for FLUX.1-dev:
>
> |                    | SDEdit | RF Edit | FlowEdit |
> |--------------------|--------|---------|----------|
> | Prompt Adherence   | 0.8083 | 0.8208  | 0.7417   |
> | Scene Preservation | 0.9708 | 0.8625  | 0.7875   |
> | Pose Preservation  | 0.8792 | 0.9167  | 0.8833   |
> | Overall Success    | 0.8708 | 0.8125  | 0.8542   |
>
> We emphasize that the study covers the complete curated set --- all 160
> edits, against every comparison method, on both backbones --- with no
> selection of image pairs or scenes. Regarding the pairwise design: as all
> methods share the same backbone (SD3 or FLUX), two or more methods can
> produce very similar results for a given case. In such cases, a
> simultaneous ranking of four outputs would be largely arbitrary for the
> similar ones, injecting noise into the aggregate rankings. We therefore
> adopted independent pairwise comparisons, which reduce each decision to a
> single binary preference. For the same reason we did not include a "tie"
> option, as it would add another layer of subjectivity (how much similarity
> constitutes a tie?) to the evaluation.
>
> **Misuse Concern:**
>
> We acknowledge that image-editing methods carry misuse potential and that
> broader-impact considerations are essential for work in this area. We note,
> however, that EDR does not improve the photorealism or believability of the
> generated content --- the capability most relevant to misuse --- which is
> inherited entirely from the publicly available pre-trained rectified-flow
> backbones (SD3, FLUX.1-dev). EDR only constrains what an edit is allowed to
> alter, narrowing rather than enlarging the space of what can be synthesized
> and reducing unrequested collateral modifications. Our full discussion,
> together with the safeguard we have implemented (default invisible
> watermarking of EDR outputs), is provided in our response to Requested
> Change2.
>
> **Requested Changes:**
>
> **1.** Our subjective (qualitative) evaluation is performed on a dataset of 80 images, for each we had two different prompts for editing resulting in 160 edits in total. The edits are done with our method as well as comparison methods on the two models (SD3 and FLUX.1-dev) based on four criteria: Prompt Adherence, Scene Preservation, Pose Preservation, and overall Success. Then our edit is compared to each of the comparison methods' output independently to avoid confusion in case of almost similar cases. For each model we executed comparison against three methods. Each comparison is done by three different participants to ensure robustness to individual preferences. Eventually the percentage of cases where our model is found superior to each comparison method is reported as the win rate. We will clarify further in the paper.

---

> ### Author Response · Authors · 2026-07-05
> **Response to Reviewer VVF8 (2/3)**
>
> **Requested Changes (Continued)**
>
> **2.** Thank you for this constructive comment. We acknowledge that image-editing methods carry misuse potential and that broader-impact considerations are essential for work in this area. It is therefore important to be precise about what EDR contributes. EDR is a training-free, inference-time procedure that is agnostic to the generative backbone, and its contribution is to improve the faithfulness of an edit — its structural and photometric consistency with the source — by suppressing unintended deviations along the editing trajectory. It does not improve the photorealism, naturalness, or believability of the generated content. That realism, which is what determines whether a manipulation can pass as authentic, is inherited entirely from the pre-trained backbone (SD3, FLUX) and is already available to anyone using these publicly released models.
> In fact, the faithfulness constraint potentially tends to make EDR outputs less convincing as standalone images than an unconstrained edit would be. As the reviewer notes in the squirrel-to-meerkat example, our method retains the structure and pose of the source rather than fully replacing the subject, so the result reads as a recognizable edit of the original squirrel rather than a wholly convincing photograph of a meerkat. EDR preserves the overall structure and color palette of the source and changes only what the prompt requires; it therefore makes edits more source-consistent, not essentially more convincing as genuine photographs, narrowing what an edit alters rather than enlarging the space of what can be synthesized. By restricting changes to those dictated by the prompt, it also reduces unrequested, collateral modifications, which could themselves be harmful, that an unconstrained editor may introduce from the backbone's priors. The capability most relevant to misuse, namely producing believable synthetic imagery, thus originates from the publicly available backbone rather than from our method.
>
> Nevertheless, as an additional safeguard, we embed an invisible blind watermark [1] in EDR outputs by default in our implementation. We have verified on our evaluation set that the watermark is imperceptible and is recovered under common post-processing, including compression, blurring, and resizing. We regard this as a provenance signal for good-faith use rather than a defense against a determined adversary, who could disable the embedding.
>
> Regarding the reviewer's suggestion to assess whether EDR-edited images remain detectable as synthetic: EDR modifies only the latent-space trajectory, and its outputs are decoded through the same pre-trained autoencoder as standard SD3 and FLUX generations. The decoder-level statistics on which synthetic-image detectors and provenance mechanisms rely are therefore unchanged, so EDR is not expected to reduce detectability relative to existing editors on the same backbones. We note this is an architectural argument rather than a measurement; we consider a systematic evaluation across detectors a valuable complement.
>
> More broadly, the realism of generated content driven by the underlying generative models rather than by editing fidelity continues to improve across the field, so reliable discrimination of synthetic content cannot rest on perceptual quality. This motivates further research on more robust methods to distinguish synthetic from real imagery, namely dedicated detectors and provenance mechanisms that do not depend on quality cues.
>
> We have added this discussion to the revised paper as a Broader Impact
> statement, with the watermark evaluation details in a dedicated appendix.

---

> ### Author Response · Authors · 2026-07-05
> **Response to Reviewer VVF8 (3/3)**
>
> **Requested Changes 2. (continued)**
>
> The added statement reads:
>
> "Broader Impact Statement:
>
> EDR enables high-fidelity, mask-free,
> prompt-driven editing of real photographs while preserving the structure
> and identity of the source image. Alongside beneficial applications in
> creative content production, photo restoration, and design, this capability
> carries dual-use risks: photorealistic manipulation of real images can
> facilitate misinformation, non-consensual edits of identifiable
> individuals, and deepfake-style content, and the strong source fidelity of
> our edits may make such manipulations harder to identify by visual
> inspection. We note that EDR builds entirely on publicly available
> pre-trained rectified flow backbones (SD3 and FLUX.1-dev) and introduces no
> new generative capability: it constrains editing trajectories of models
> already broadly accessible through existing editors, restricting what an
> edit may alter rather than enlarging the space of what can be synthesized,
> so the risk it adds is incremental over already-available editors. As
> mitigations, our implementation embeds an invisible blind watermark in EDR
> outputs by default, serving as a provenance signal for good-faith use, and our released code will be accompanied by
> usage guidelines requiring compliance with the acceptable-use policies of
> the underlying models, which prohibit deceptive manipulation of real
> individuals. Moreover, because EDR outputs are decoded through the same
> pre-trained autoencoders as standard SD3 and FLUX generations, existing
> synthetic-image detectors and provenance mechanisms developed for these
> models (e.g., content-credential standards such as C2PA) remain applicable.
> This work motivates continued research on detecting AI-edited imagery."
>
> **Broader Impact Concerns:**
> Please see our response to Requested changes 2, where includs the discussion of misuse risks, the default watermarking
> safeguard we have implemented and verified, and the Broader Impact statement added to the revised paper.
>
> We are grateful for the reviewer's careful and thorough assessment, which has substantively strengthened the paper — in particular its treatment of safeguards and responsible use. We hope our responses and the accompanying revisions address the concerns raised, and we welcome any further discussion, particularly regarding the added Broader Impact statement.
>
> **References:**
>
> [1] Zhang, Kevin Alex and Xu, Lei and Cuesta-Infante, Alfredo and Veeramachaneni, Kalyan. Robust Invisible Video Watermarking with Attention. MIT EECS, September 2019

---

### Review · Reviewer_HJhe · 2026-06-17

**Summary Of Contributions:**

This paper addresses the challenge of maintaining faithfulness to the source image during generative-prior-based image editing without requiring computationally heavy test-time optimization. To achieve this, the authors introduce Editing via Degraded Representations (EDR), an optimization-free framework that projects editing trajectories onto a tailored degraded space combining Gaussian spatial smoothing with dynamic range reduction to suppress unintended deviations. Evaluations on models like Stable Diffusion 3 and FLUX.1-dev demonstrate that the proposed method preserves foundational image characteristics while successfully accommodating the requested semantic modifications.

**Audience:**

Yes

**Audience Explanation:**

The findings of this paper are relevant to TMLR's audience because they present an optimization-free trajectory-control framework that addresses structural and color preservation challenges in rectified flow and diffusion-based image editing.

**Claims And Evidence:**

Yes

**Claims Explanation:**

The claims in the submission are supported by a combination of quantitative experiments using standard metrics, qualitative comparisons across multiple generative backbones, a user study, and systematic ablation analysis.

**Requested Changes:**

The combination of Gaussian blur and dynamic range reduction may fail to preserve highly complex high-frequency textures that simple spatial smoothing cannot handle.

The quantitative evaluation is performed on a relatively small dataset of 80 images, which may not fully reflect the real-world robustness of the framework.

The transition schedule between the projected trajectory and the native trajectory relies on a hand-crafted decay factor that might require per-task tuning.

There is no detailed discussion or quantitative measurement of the additional inference latency introduced by calculating the degradation and projection at each step.

The manuscript does not present failure cases or analyze the explicit performance boundaries of the proposed EDR method.

---

> ### Author Response · Authors · 2026-07-05
> **Response to Reviewer HJhe (1/2)**
>
> We thank the reviewer for the constructive feedback and respond to each point below. All changes described are included in the revised manuscript.
>
> **Requested Changes:**
>
> **1.** We agree that an idealized restriction to a smoothed representation could not, by itself, synthesize complex high-frequency content. However, EDR does not confine the trajectory to the degraded space. At each step, the update preserves in full the component of the direct velocity aligned with the degraded direction and attenuates only the orthogonal remainder, and this attenuation follows a decaying schedule: it is strongest near the source, where structural fidelity matters most, and is progressively released so that the final steps follow the model's native velocity $V^{dir}_t$ essentially unconstrained. High-frequency detail is therefore synthesized by the backbone in the later portion of the trajectory, as in standard generation, while the early constraint prevents it from drifting away from the source structure. This is corroborated empirically: our results include edits requiring complex fine-grained textures — e.g., animal fur and coat patterns (Figures 5 and A) and full style transfer (Figure B) — which EDR renders correctly while preserving source structure.
>
> **2.** We thank the reviewer for raising this point. Our evaluation rests on two complementary datasets. The quantitative comparison (Tables 3 and 4) is executed on the standard benchmark and prompts of [1]— the same protocol under which the state-of-the-art methods we compare against report their results — which contains diverse real images with multiple target prompts per image, ensuring both breadth and direct comparability with previously published numbers. Complementing this, the 80-image set was manually curated for the qualitative evaluation and the user study: each image is edited with two distinct target prompts, executed on both SD3 and FLUX.1-dev, and every edit is judged against each comparison method in an independent pairwise trial by three participants, across four criteria (Prompt Adherence, Scene Preservation, Pose Preservation, and Overall Success).
> Regarding whether this scale reflects real-world robustness: the images are sourced from LSDIR, DIV2K, and [1], and the edit types span object replacement, color, text, and style. Most importantly, the automated metrics and the human evaluation are conducted on different image sets under different protocols, yet yield the same conclusion — EDR consistently leads on scene and pose preservation, the criteria most representative of source-faithful editing, while maintaining prompt adherence competitive with state-of-the-art methods. This agreement across independent datasets and evaluation modalities is, we believe, stronger evidence of robustness than either evaluation would provide alone. That said, we agree that broader benchmarking is always valuable. We have clarified the respective roles of the two datasets in Section 4.1.3.
>
> **3.** Although per-case tuning of the decay factor could further improve individual results, all qualitative and quantitative experiments in the paper use a single fixed configuration per backbone (Section 4.1.1), applied uniformly across all images, prompts, and edit types — including object replacement, color, text, and style edits — with no per-task adjustment. Moreover, the hyperparameter study in Appendix B shows that neighboring configurations form a smooth trade-off curve rather than a narrow optimum, indicating graceful degradation around the selected setting. Together these indicate that EDR does not rely on per-task tuning.
>
> **4.** We thank the reviewer for this constructive comment. The degradation and projection introduce no additional network evaluations. Each step requires the same two velocity-field passes as the underlying direct-path formulation, and the added operations (a separable Gaussian smoothing and a vector projection) are $O(N)$ in the latent dimension, which is orders of magnitude below the cost of a single backbone forward pass. We have measured the per-image execution time for our method as well as comparison method. The results are as following:
>
> Efficiency Results on SD3:
>
> |                     |       EDR       | FlowEdit ($n_{avg}=2)$ |  ODE Inversion  |
> |---------------------|:---------------:|:--------------------:|:---------------:|
> | Wall Clock Time (s) | 13.6 $\pm$ 0.31 |   18.25 $\pm$ 1.26   | 6.47 $\pm$ 0.33 |
> | Peak Memory (GB)    |        19       |          19          |        19       |
>
> Efficiency Results on FLUX.1-dev
>
> |                     |      EDR      | FlowEdit ($n_{avg}=2)$ | RF Edit |
> |---------------------|:-------------:|:--------------------:|:-------:|
> | Wall Clock Time (s) | 23.94$\pm$.63 |    44.59$\pm$0.24    |  55.99  |
> | Peak Memory (GB)    |       33      |          33          |    35   |

---

> ### Author Response · Authors · 2026-07-05
> **Response to Reviewer HJhe (2/2)**
>
> **5.** A common limitation shared by faithfulness-oriented editing methods is difficulty with drastic semantic edits, where the source and target share little structure — e.g., editing a cat into an elephant — and EDR inherits this: when the requested edit conflicts with structural preservation, the constraint resolves toward fidelity, yielding an under-edited output rather than an unfaithful one. This failure mode is the boundary of the fidelity–editability trade-off that motivates our design, but it does bound the method's applicability to edits that retain a structural relation between source and target. We have added a Limitations section to the appendix characterizing this failure mode and the scope of the method.
>
> The limitation section reads:
>
> Limitations:
>
> EDR is designed for faithful editing, and its principal limitation is the
> boundary of that objective: drastic semantic edits in which the source and
> target share little structural correspondence. When the target prompt
> requires large changes in shape, category, or layout --- for example,
> transforming a cat into an elephant, or replacing an object with one of a
> substantially different geometry --- the trajectory constraint, which
> suppresses deviations from the source structure most strongly in the early
> editing steps, opposes the very modifications the prompt demands. In such
> cases the method resolves the conflict toward fidelity: the characteristic
> failure mode is an under-edited output that remains close to the source
> rather than an unfaithful or corrupted one. Relatedly, since the degraded
> representation is built from spatial smoothing, edits whose primary content
> is fine-grained and spatially localized in the early trajectory can require
> a larger editing strength $t_0$ to fully materialize. We regard this
> behavior as the intended operating point of the fidelity--editability
> trade-off --- faithful editing is well-posed only when a meaningful
> structural relation between source and target exists --- but it delimits
> the scope of EDR: for creative transformations that deliberately discard
> source structure, unconstrained editors or pure generation are the more
> suitable tools. Extending the degraded representation to adapt its
> invariances to the magnitude of the requested edit is an interesting
> direction for future work.
>
> We thank the reviewer for the helpful suggestions, and remain available for any further questions during the discussion period.
>
> [1] Kulikov, Vladimir, et al. "Flowedit: Inversion-free text-based editing using pre-trained flow models." Proceedings of the IEEE/CVF International Conference on Computer Vision.

---

### Review · Reviewer_6Jib · 2026-06-23

**Summary Of Contributions:**

This paper introduces Editing via Degraded Representations (EDR), a training-, inversion- and optimization-free text-driven image editing method on rectified-flow (RF) backbones. It extends FlowEdit's "direct path" formulation, where the editing direction at each step is the difference of the target-prompt and source-prompt velocity fields. The main addition is to constrain this editing trajectory: at each step the editing direction is projected onto a degraded version of itself, where the degradation is a combination of Gaussian spatial smoothing and a dynamic-range reduction. The intuition is that projecting onto a representation that is invariant to a given characteristic removes the editing components along that characteristic and thus removes "secondary" unintended modifications while preserving the "primary" prompt-driven edit. The source fiedlity is traded off against output quality using a time-dependent blend between the projected direction and the original editing direction along with an editing-strength parameter.

The contribution is evaluated with: (1) a user study (30 participants) of win rates over SDEdit, ODE Inversion, RF Edit, and FlowEdit across Prompt Adherence, Scene Preservation, Pose Preservation, and Overall Success; (2) quantitative results on the FlowEdit benchmark using CLIP-text, CLIP-image, DINO, LPIPS, and DreamSim; (3) an ablation removing each of the two degradation operators ; and (4) also a hyperparameter Pareto-front grid search plus full pseudocode.

Strengths:
* The main idea is quite clean and intuituve and also a novel perspective in the RF-editing literature. I think that the Fig 2 conveys this well.
* The method is simple and lightweight.
* Source-fidelity results are consistent and also clearly positive.
* The ablation cleanly isolates the two operators and shows both are needed.

Weaknesses
* I think that many claims are overstated as compared to the evidence. Example: a "theoretical analysis" / "principled" method and a claim to "establish the existence of degraded representations" that the methodology section does not actually provide; They say “agnostic to the underlying generative backbone" despite only RF experiments; and a "new state-of-the-art" framing that understates a real fidelity-versus-editability trade-off.
* There is also an apparent inconsistency between how the decay schedule is described in words and how it actually behaves given the order in which the method runs.
* There is a conceptual underspecification of what it means to apply image-style degradations (an intensity remapping, a spatial blur) to rectified-flow latents and velocities and also a mismatch between the multi-dimensional "subspace" intuition of Figure 2 and the single-direction projection the authors actually used.
* No efficiency or runtime comparison despite efficiency being a stated motivation; baseline numbers imported from another paper rather than re-run (though this is a minor concern and is encouraged but not really necessary); no error bars or significance testing; thin user-study methodology.

**Audience:**

Yes

**Audience Explanation:**

This is an active and well-populated research area. The fidelity improvements are consistent and the mechanism is simple enough to drop onto an existing direct-path editor so it is of practical interest to practitioners and conceptually interesting to researchers working on editing and flow control.

**Broader Impact Concerns:**

The submission has no Broader Impact / ethics statement. The method produces high-fidelity, mask-free, prompt-driven edits of real photographs and the paper explicitly emphasizes real-image editing with strong structure and identity preservation. This carries clear dual-use risk: photorealistic manipulation of real images enables misinformation, non-consensual edits of identifiable people and deepfake-style content, and the strong source-fidelity property can make such manipulations harder to detect. I think that the authors should add a short Broader Impact Statement that acknowledges these misuse risks and discusses possible mitigations.

**Claims And Evidence:**

No

**Claims Explanation:**

I think the empirical claim that EDR improves source fidelity is well supported. The user study, quantitative tables and ablation are consistent and also mutually corroborating. However, a lot of the paper's central claims are not adequately supported as written. This is the main reason why I answer "No" at this stage though I think most of these are fixable during discussion.

1. The authors do not deliver the theoretical / "principled" claims. The abstract and contributions say the authors "establish the existence of such degraded representations" and provide a "theoretical analysis" and a "principled method." The methodology section has no theorem, existence result or guarantee that the projection preserves desired content or removes unintended modifications. It is a heuristic projection.
2. As I also mentioned earlier, there is a concrete inconsistency between the prose and the actual behavior of the core schedule. The text says the method prioritizes the projected direction "during the initial timesteps" and gradually shifts toward the original editing direction at later steps. But given the blending coefficient used and the order in which the algorithm runs (from the editing-strength start point down toward the final image), the projection is in fact weakest at the start and strongest at the end. For the FLUX settings this is extreme: the projection contributes essentially nothing at the first step. Calling the parameter a "decay rate" also adds to the confusion since the projection's influence actually increases along the trajectory.
3. "Agnostic to the underlying generative backbone" is also not demonstrated in the paper. The whole formulation and every experiment are rectified-flow specific. No diffusion or non-RF backbone is shown. I think that this should be rescoped to "applicable across RF backbones" unless backbone-agnosticism is actually demonstrated.
4. "New state-of-the-art" understates a fidelity-versus-editability trade-off. The gains are concentrated in faithfulness and not prompt adherence. On FLUX, EDR's CLIP-text score is slightly below FlowEdit's and the user study shows prompt-adherence win rates of essentially a tie. This is a legitimate and useful operating point but the SOTA framing should acknowledge that EDR trades a little editing strength for large fidelity gains rather than implying uniform superiority.
5. Also a major issue is that efficiency is motivated but never measured. The paper repeatedly argues against the cost of test-time optimization and against FlowEdit's aggregation of multiple model predictions per step but provides no runtime, function-evaluation or memory comparison.
6. The degradation operator is conceptually underspecified. It is described in image term but is applied to rectified-flow latents and to finite-difference velocities whose values are not bounded the way pixel intensities are. It is unclear over what grid the blur operates and why an intensity remap is meaningful on a velocity. The projection is also onto a single direction, which collapses the editing direction to a scalar multiple of the degraded one. Now this is a much more aggressive operation than the subspace/plane projection suggested by Figure 2 which moreover leans on an edge representation that the final method never uses. This gap between the illustrative story and the actual operation should be reconciled.

**Requested Changes:**

Critical:
1. Please resolve the decay-schedule inconsistency and reconcile the prose description with how the blending coefficient actually behaves given the order in which the algorithm runs. Correct whichever of the description, the schedule or the time convention is wrong and state unambiguously how the projection's influence varies along the trajectory. Clarify the meaning of "decay rate" given that the projection's contribution actually increases toward the end.
2. Tone down or substantiate the theoretical claims and remove "establish the existence," "theoretical analysis," and "principled" from the abstract and introduction or if possible add an actual analytical result (for instance, conditions under which the projection provably suppresses a targeted characteristic) (The latter option is preferred if possible). Otherwise present the method honestly as a heuristic construction.
3. You will have to clarify the degradation operator on latents and velocities. Define the grid over which the Gaussian blur acts, justify applying an intensity-range reduction to velocity values and reconcile the single-direction projection with the subspace intuition of Figure 2. Please also state explicitly that the deployed method uses only the blur and range-reduction and the edge representation is illustrative and unused.
4. Report wall-clock time and/or number of function evaluations and memory for EDR versus FlowEdit, RF Edit, RF Inversion and at least one optimization-based method for efficiency comparison since it is a central motivation of the work.
5. Either re-run baselines under a single unified protocol (if possible though can be ignored) or you can also clearly report variability (multiple seeds / error bars) for the quantitative metrics and report statistical detail and significance for the user study so the results are ensured to be an apples-to-apples comparison.

The below would help in strengthening the work but is not critical:
1. Reframe the SOTA / editability claim to acknowledge the fidelity-versus-prompt-adherence trade-off and explicitly note the FLUX CLIP-text result and the roughly 50% prompt-adherence win rates.
2. Broaden the quantitative baselines to include attention-injection and other optimization-free editors the paper discusses so the comparison is not limited to three or four methods.
3. Expand the user-study description and please include protocol (pairwise versus simultaneous), how rankings are aggregated into win rates, tie handling, number of comparisons per participant, inter-rater agreement and confidence intervals. Also report quantitative metrics on the curated 80-image / 160-prompt set.
4. Demonstrate or explicitly scope the "backbone-agnostic" claim ideally with at least one non-RF/diffusion result otherwise rescope it to RF backbones.
5. Fix minor clarity issues: a missing equals sign in the definition of the source velocity field; the ambiguous definition of the finite-difference velocity at the first step (there is no previous state at the first iteration).

---

> ### Author Response · Authors · 2026-07-05
> **Response to Reviewer 6Jib (1/5)**
>
> We thank the reviewer for their comprehensive and careful review. We addressed the concerns point by point below, and the revised manuscript incorporates the corresponding changes.
>
> **Weaknesses:**
>
> **W1.**
> We rewrote the claims to match the evidence. Specifically, in the abstract, we replaced "that is, in principle, agnostic to the underlying backbone." (line 9 and 10), with "for rectified flow models, such as SD3 and FLUX.1-dev." Also, on the line 12 of the abstract, we replaced "We first establish the existence of such degraded representations" with "We first illustrate the potential of such degraded representations", and on 13th line we deleted the word "principled".  Finally, in the last two lines of the abstract, we replaced "EDR achieves precise, high-quality edits with superior fidelity," with " EDR achieves precise, high quality edits with superior fidelity with negligible restriction in editability,". The revised abstract reads:
>
> Abstract:
>
> Rectified flow and diffusion-based models currently represent the state-of-the-art in image editing, leveraging powerful pre-trained generative priors to produce visually compelling modifications. Despite their impressive capabilities, maintaining faithfulness to the source image -- preserving structure and photometric characteristics while satisfying a target prompt -- remains a persistent challenge in this domain. Direct traversal between source and target distributions in rectified flow frameworks offers a promising direction for improving fidelity. However, identifying trajectories that are both semantically effective and strictly structure-preserving remains an open problem.  In this work, we propose an optimization- and inversion-free image editing framework for rectified flow models, such as SD3 and FLUX.1-dev. Our central insight is to operate within a carefully designed degraded representation space that constrains editing trajectories and suppresses unintended collateral modifications to the target. We first illustrate the potential of such degraded representations for generative-prior-based editing and then develop a method to project editing trajectories onto this space. The resulting method, Editing via Degraded Representations (EDR), systematically eliminates unfaithful trajectory deviations while preserving the flexibility required to satisfy the target text prompt. Extensive quantitative and qualitative evaluations demonstrate that EDR achieves precise, high quality edits that offers superior fidelity with negligible restriction in editability, establishing a new state-of-the-art in faithful image editing. Code will be released upon acceptance.
>
> In the contribution part, in first bullet, we deleted the word "principled", and in the third bullet we replaced "maintaining exceptional source fidelity", with "maintaining remarkable source fidelity with negligible restriction in editability". Additionally, the opening of Section 3 no longer refers to a `theoretical analysis'.
>
> **W2.**
> We appreciate the reviewer's careful examination of the equation and its accompanying explanation. We have resolved it in Requested Changes Critical 1.
>
> **W3.**
> The degradation operators act directly on the latent velocity and are used only to remove components of the editing update, by projecting the velocity onto the degraded direction. In this sense the mechanism constrains how much the latent is allowed to change, rather than transforming the latent to force a prescribed change in pixel space. Because the latent spaces of SD3 and FLUX.1-dev are spatially aligned with the decoded image, a spatial smoothing operator applied across the latent grid acts analogously to smoothing in image space and suppresses local structural variation, and the range-reduction operator similarly constrains latent magnitude. We do not claim exact equivalence between these latent-space operators and their pixel-space counterparts. The degraded representation is used to define a direction whose suppression removes structural and photometric drift, and its effectiveness is verified empirically by the ablation study.
>
> With respect to Figure 2, it is presented for the sake of clarity of how degraded representations are invariant to specific characteristics, which motivates editing through degraded representations. However, it does not directly represent our method, especially the edge representation. We will explicitly mention this in revised paper Figure 2 caption.

---

> ### Author Response · Authors · 2026-07-05
> **Response to Reveiwer 6Jib (2/5)**
>
> **W4.**
> Thank you for highlighting this point. We measured wall-clock time and peak GPU memory for our method, the comparison methods, and one optimization-based method (Imagic [1]). EDR runs faster than FlowEdit on both backbones and than RF Edit on FLUX with equal or lower peak memory, while Imagic requires almost five minutes per image despite using a smaller backbone. The full results and discussion are presented in Requested Changes Critical 4.
>
> Regarding error bars and significance testing, the baseline quantitative results are adopted from [2], which does not provide this information (please see Requested Changes Critical 5). The expanded user-study description, including inter-rater agreement, is provided in our response to point 3 of the additional requested changes.
>
> **Claims:**
>
> **Claims1.**
> We have rewritten the claims to match the evidence: the revised abstract and contributions no longer use "establish the existence"
> or ``principled'', and the opening of Section~3 no longer refers to a "theoretical analysis". Please see our response to W1.
>
> **Claims2.**
> We thank the reviewer for raising this point.The inconsistency was a transcription error in Eq.11: the correct schedule is $\alpha = t^{\gamma}$, under which the projection weight is largest at the first editing step and decays monotonically to zero at the final step, matching the prose. This is also consistent with the "decay rate" naming, as larger $\gamma$ yields faster decay of the projection weight along the trajectory. Please see Requested Changes Critical 1 for details.
>
> **Claims3.**
> Our method follows the rectified-flow formulation, and by "model-agnostic" we referred to rectified-flow backbones such as SD3 and FLUX.1-dev. We have clarified this scope in the revised version (please see the response to Weakness 1).
>
> **Claims4.**
> Because our method restricts the editing trajectory to improve fidelity, a small reduction in editing strength is inherent to the approach. As the reviewer notes, this is concentrated in prompt adherence, where EDR is essentially comparable with FlowEdit (CLIP_txt 0.335 vs 0.337 on FLUX and comparable user-study win rates), while the fidelity gains are substantial (DINO 0.739 vs 0.682, LPIPS 0.170 vs 0.223 on FLUX). We have reframed the abstract and introduction around this fidelity–editability balance rather than implying uniform superiority (please see our response to Weakness 1).
>
> **Claims5.**
> We have now measured efficiency directly. We report wall-clock time and peak GPU memory for EDR and the comparison methods on the first five edits of the quantitative dataset. On SD3, EDR requires 13.6 s per edit versus 18.25 s for FlowEdit ($n_{avg}=2$), and on FLUX, 23.94 s versus 44.59 s for FlowEdit and 55.99 s for RF Edit, with equal or lower peak memory in all cases. This is consistent with EDR requiring no aggregation of multiple model predictions per step. We additionally benchmarked Imagic as a representative optimization-based method; despite using a smaller backbone (SD v1.5), it requires almost five minutes per image, confirming that per-image optimization is not competitive with optimization-free methods in runtime. The full tables are provided in our response to Requested Changes Critical 4.
>
> **Claims6.**
> We have clarified the operator and its scope in the revision. The degradation $\mathcal{D}_c$ acts on the finite-difference velocity, i.e., the temporal difference of the latent states, and the Gaussian blur operates over the 2D spatial grid of the latent representation, applied per channel with kernel size 30 and $\sigma = 5$. Because the latent spaces of SD3 and FLUX.1-dev are spatially aligned with the decoded image, smoothing over the latent grid suppresses local structural variation analogously to image-space blurring; we do not claim exact equivalence between the latent-space operators and their pixel-space counterparts. The dynamic range reduction bounds the magnitude of velocity values, suppressing the rapid latent shifts responsible for incidental intensity and color drift without restricting the structural transitions required by the edit.
>
>  Regarding Figure 2, it illustrates an idealized subspace projection; deriving an exact orthogonal projection for these characteristics is computationally intractable at inference time, so Eq. 8 employs a single-direction projection onto $\tilde{V}_t$ as a computationally efficient surrogate. We now state explicitly in Section 3.1 that the edge representation is purely illustrative and that the deployed method uses only Gaussian structural suppression and dynamic range reduction. For further discussion, please see our response to Requested Changes-Critical-3.

---

> ### Author Response · Authors · 2026-07-05
> **Response to Reveiwer 6Jib (3/5)**
>
> **Requested Changes:**
>
> **Critical:**
>
> **Critical1.**
> We thank the reviewer for raising this point. The inconsistency is in the transcription of Eq. 11. The implemented and intended equation is $\alpha = t^{\gamma}$, resulting in a projection weight that is largest at the first editing step and decreases to 0 at the final step, as the editing trajectory runs from $t_0$ toward 0. The projection is thus weighted most heavily near the source and relaxed toward the output, and the parameter $\gamma$ acts as a decay rate on the projection weight. In the revised version, we have updated Eq. 11 and Line 12 of Algorithm A to match.
>
> **Critical2.**
> Thank you for this comment. By construction, the projection preserves
>     the component of the editing velocity aligned with the degraded
>     direction and only restricts, rather than removes, the remaining
>     directions. Accordingly, we have toned down the claims: the revised
>     abstract and contributions no longer use ``establish the existence''
>     or principled (please see W1), and Section 3 no longer
>     refers to a theoretical analysis.
>
> **Critical3.**
> The degradation function $D_c$ operates on the temporal difference of the latent variables. The Gaussian blur $G_{\sigma}$ acts over the 2D spatial grid of these latent representations. Specifically, the convolution is applied spatially across the latent channels using a kernel size of 30 and $\sigma=5$, as detailed in our experimental setup.
>
> Applying dynamic range reduction to the velocity bounds the update magnitudes. It specifically induces invariance to incidental intensity and color shifts that frequently manifest as secondary, unintended modifications during the editing trajectory. By squashing these velocity values, we actively suppress the rapid latent shifts responsible for spurious color deviations without restricting necessary structural transitions.
>
> Figure 2 illustrates the concept of projecting out orthogonal characteristics (i.e., a true subspace projection). However, deriving an exact orthogonal projection matrix for these highly non-linear generative characteristics is computationally intractable at inference time. Therefore, Equation 8 approximates this by computing a single-direction projection of the direct velocity $V^{dir}_t$ onto the degraded velocity vector $\tilde{V}_t$. We will update Section 3.1 to explicitly state that this 1D projection acts as a computationally efficient surrogate, filtering out components of the editing trajectory that do not align with the structurally smoothed and range-reduced path.
>
> We have added an explicit statement to Section 3.1 clarifying that the edge representation is purely illustrative. As detailed in Section 3.2 and the ablation studies, our deployed Editing via Degraded Representations (EDR) method strictly relies on Gaussian structural suppression and dynamic range reduction; the edge representation is not used in the deployed framework.
>
> **Critical4.**
> We have measured wall-clock time and peak GPU memory for our method and the comparison methods on the first five edits of the quantitative dataset, and the results are reported below. EDR requires less runtime than FlowEdit on both backbones (13.6 s vs. 18.25 s on SD3; 23.94 s vs. 44.59 s on FLUX) and than RF Edit on FLUX (55.99 s), with equal or lower peak memory in all cases, consistent with EDR requiring no aggregation of multiple model predictions per step. Among the compared methods, ODE inversion attains the lowest runtime on SD3 (6.47 s), but this speed comes at a substantial fidelity cost: in the user study, EDR wins against ODE inversion on all four criteria, including a 93\% overall-success win rate. EDR therefore offers the lowest runtime among methods operating at state-of-the-art fidelity.
>
> As requested, we also executed the same experiment for Imagic. The much longer execution time — almost five minutes per image despite the relatively small backbone (SD V1.5) — confirms that optimization-based methods are not competitive with optimization-free methods like ours in terms of time efficiency.
>
> Efficiency Results on SD3:
>
> |                     |       EDR       | FlowEdit ($n_{avg}=2)$ |  ODE Inversion  |
> |---------------------|:---------------:|:--------------------:|:---------------:|
> | Wall Clock Time (s) | 13.6 $\pm$ 0.31 |   18.25 $\pm$ 1.26   | 6.47 $\pm$ 0.33 |
> | Peak Memory (GB)    |        19       |          19          |        19       |
>
> Efficiency Results on FLUX.1-dev
>
> |                     |      EDR      | FlowEdit ($n_{avg}=2)$ | RF Edit |
> |---------------------|:-------------:|:--------------------:|:-------:|
> | Wall Clock Time (s) | 23.94$\pm$.63 |    44.59$\pm$0.24    |  55.99 $\pm$0.12  |
> | Peak Memory (GB)    |       33      |          33          |    35   |
>
> Efficiency Results (Imagic)
>
> | Wall Clock Time (s) | 299.4+0.8 |
> |---------------------|-----------|
> | Peak Memory (GB)    | 17        |

---

> ### Author Response · Authors · 2026-07-05
> **Response to Reveiwer 6Jib (4/5)**
>
> **Critical5.**
>
> For the quantitative metrics, the baseline results are adopted from [2], so that all methods are compared under the settings and numbers reported by the original authors; that source does not provide variability information, and, as the reviewer notes, re-running all baselines under a unified protocol can be set aside. The expanded protocol description and inter-rater agreement tables are provided in our response to point 3 of the additional requested changes.
>
> **Non-critical:**
>
> **Non-critical1:**
>
> We have reframed the claims around the fidelity–editability balance rather than uniform superiority: the abstract and contributions now state that EDR achieves superior fidelity with negligible restriction in editability (the revised text is provided in our response to Weakness 1). Additionally, we have added an explicit note to the experimental section acknowledging that on FLUX, EDR's $\text{CLIP}_{\text{txt}}$ is marginally below FlowEdit's (0.335 vs. 0.337) and the prompt-adherence win rates against RF Edit and FlowEdit are approximately comparable, while the fidelity gains are substantial.
>
> **Non-critical2:**
>  We appreciate the reviewer's constructive suggestion. Our comparison focuses on trajectory- and inversion-based editors, which are the most methodologically comparable to EDR's trajectory constraint mechanism. Attention-injection methods operate through a different control mechanism (feature/attention manipulation). We agree that broadening to attention-injection editors is a worthwhile extension and a useful direction for broader benchmarking.
>
> **Non-critical3:**
> We thank the reviewer for their constructive feedback.  We have expanded the user-study description in the revised paper as follows.
>
> The study covers 160 test cases (80 images × 2 target prompts) per backbone (SD3 and FLUX.1-dev). For each case, EDR's output is compared against each comparison method's output in an independent pairwise trial, rather than a simultaneous ranking of all methods; we adopted this design to avoid confusion in cases where several outputs are nearly similar. In each trial, participants are shown the source image, the target prompt, and the two anonymized outputs in randomized order, and indicate the superior output for each of the four criteria (Prompt Adherence, Scene Preservation, Pose Preservation, Overall Success). The choice is a forced binary decision. Each pairwise trial is judged by three different participants to ensure robustness to individual preferences.
>
> The win rates in Tables 1 and 2 are the percentage of individual judgments in which EDR is preferred.
>
> For inter-rater reliability, we report the mean pairwise raw agreement: for each trial and criterion, the fraction of the three rater pairs giving the same preference, averaged over all trials and reported per criterion and per baseline below. Agreement ranges from 0.74 to 0.97 across criteria and baselines (chance level for a binary preference is 0.5), indicating substantial consistency.
>
> Mean Pairwise Raw Agreement for SD3:
>
> |                    | SDEdit | ODE Inv. | FlowEdit |
> |--------------------|--------|----------|----------|
> | Prompt Adherence   | 0.8542 | 0.8583   | 0.8250   |
> | Scene Preservation | 0.9333 | 0.8958   | 0.8333   |
> | Pose Preservation  | 0.9000 | 0.8917   | 0.8458   |
> | Overall Success    | 0.8292 | 0.8500   | 0.8625   |
>
> Mean Pairwise Raw Agreement for FLUX.1-dev:
>
> |                    | SDEdit | RF Edit | FlowEdit |
> |--------------------|--------|---------|----------|
> | Prompt Adherence   | 0.8083 | 0.8208  | 0.7417   |
> | Scene Preservation | 0.9708 | 0.8625  | 0.7875   |
> | Pose Preservation  | 0.8792 | 0.9167  | 0.8833   |
> | Overall Success    | 0.8708 | 0.8125  | 0.8542   |
>
> **Non-critical 4.**
>     We have rescoped the claim to rectified-flow backbones, as the
>     reviewer suggests: the abstract now reads "an optimization- and
>     inversion-free image editing framework for rectified flow models,
>     such as SD3 and FLUX.1-dev" in place of the backbone-agnostic
>     phrasing (please see our response to W1 for the full revised
>     text). Within this scope, the claim is supported by the paper's
>     experiments on two distinct large-scale RF backbones.
>
> **Non-Critical5.**
> Thank you for highlighting the minor issues. We have fixed these in the revised version of the paper.

---

> ### Author Response · Authors · 2026-07-05
> **Response to Reveiwer 6Jib (5/5)**
>
> **Broader Impact Concern:**
>
> We agree that a Broader Impact statement is warranted and have added one to the revised paper. It acknowledges the dual-use risks the reviewer identifies and discusses mitigations. The added statement reads:
>
> Broader Impact Statement:
>
> EDR enables high-fidelity, mask-free, prompt-driven editing of real photographs while preserving the structure and identity of the source image. Alongside beneficial applications in creative content production, photo restoration, and design, this capability carries dual-use risks: photorealistic manipulation of real images can facilitate misinformation, non-consensual edits of identifiable individuals, and deepfake-style content, and the strong source fidelity of our edits may make such manipulations harder to identify by visual inspection. We note that EDR builds entirely on publicly available pre-trained rectified flow backbones (SD3 and FLUX.1-dev) and introduces no new generative capability: it constrains editing trajectories of models already broadly accessible through existing editors, restricting what an edit may alter rather than enlarging the space of what can be synthesized, so the risk it adds is incremental over already-available editors. As mitigations, our implementation embeds an invisible blind watermark in EDR outputs by default, serving as a provenance signal for good-faith use, and our released code will be accompanied by usage guidelines requiring compliance with the acceptable-use policies of the underlying models, which prohibit deceptive manipulation of real individuals. Moreover, because EDR outputs are decoded through the same pre-trained autoencoders as standard SD3 and FLUX generations, existing synthetic-image detectors and provenance mechanisms developed for these models (e.g., content-credential standards such as C2PA) remain applicable. This work motivates continued research on detecting AI-edited imagery.
>
> We thank the reviewer again for the careful and constructive review, and would be happy to address any remaining concerns.
>
> References:
> [1] Kawar, Bahjat, et al. "Imagic: Text-based real image editing with diffusion models." CVPR 2023.
>
> [2] Kulikov, Vladimir, et al. "Flowedit: Inversion-free text-based editing using pre-trained flow models." ICCV 2025.